# MAPK4 promotes triple negative breast cancer growth and reduces tumor sensitivity to PI3K blockade

Wei Wang[1,6], Dong Han[1,6], Qinbo Cai [1], Tao Shen[1], Bingning Dong[1,2], Michael T. Lewis[1,3], Runsheng Wang[1], Yanling Meng[1,4], Wolong Zhou[1], Ping Yi[1], Chad J. Creighton [2,3], David D. Moore [1,5] & Feng Yang [1✉]

About 15–20% of breast cancer (BCa) is triple-negative BCa (TNBC), a devastating disease with limited therapeutic options. Aberrations in the PI3K/PTEN signaling pathway are common in TNBC. However, the therapeutic impact of PI3K inhibitors in TNBC has been limited and the mechanism(s) underlying this lack of efficacy remain elusive. Here, we demonstrate that a large subset of TNBC expresses significant levels of MAPK4, and this expression is critical for driving AKT activation independent of PI3K and promoting TNBC cell and xenograft growth. The ability of MAPK4 to bypass PI3K for AKT activation potentially provides a direct mechanism regulating tumor sensitivity to PI3K inhibition. Accordingly, repressing MAPK4 greatly sensitizes TNBC cells and xenografts to PI3K blockade. Alto-gether, we conclude that high MAPK4 expression defines a large subset or subtype of TNBC responsive to MAPK4 blockade. Targeting MAPK4 in this subset/subtype of TNBC both represses growth and sensitizes tumors to PI3K blockade.

[1] Department of Molecular and Cellular Biology, Baylor College of Medicine, Houston, TX, USA. [2] Department of Medicine, Baylor College of Medicine, Houston, TX, USA. [3] Dan L Duncan Comprehensive Cancer Center, Baylor College of Medicine, Houston, TX, USA. [4] Adrienne Helis Malvin Medical Research Foundation, New Orleans, LA, USA. [5] Present address: Nutritional Sciences and Toxicology, University of California – Berkeley, Berkeley, CA, USA. [6] These authors contributed equally: Wei Wang, Dong Han. ✉email: fyang@bcm.edu

The AKT/mTOR signaling pathway plays critical roles in regulating cell proliferation, survival, and metabolism. In the canonical pathway, PI3 kinase (PI3K) catalyzes phosphatidylinositol-3,4,5-triphosphate (PIP3) production, and PIP3 binds and recruits AKT to the plasma membrane for activation[1–3]. Phosphorylation of AKT at both Thr308 (T308) and Ser473 (S473) is requisite for full AKT activation. mTOR forms two different complexes, mTORC1 and mTORC2. AKT activates mTORC1, which integrates extracellular stimuli and nutrient signals to modulate cell growth, autophagy, and metabolism[4]. In contrast, mTORC2 is the main S473 kinase of AKT[5], while PDK1 is the major T308 kinase[6]. The PI3K/AKT/mTOR signaling pathway is crucial in regulating tissue home-ostasis, and its dysregulation can cause various pathological conditions including cancers.

About 15–20% of all breast cancers (BCa) are triple-negative (TNBC)[7–9], a devastating disease with limited therapeutic options. Aberrations in the PI3K/PTEN/AKT signaling pathway are common in TNBC[10]. The FDA has recently approved the α-isoform-specific PI3K inhibitor Piqray (Alpelisib) to be used in combination with fulvestrant to treat hormone receptor-positive, HER2-negative, PIK3CA-mutated, advanced or metastatic BCa following progression on or after an endocrine-based regimen. However, PI3K inhibitors have shown only limited therapeutic responses when use to treat TNBC[11,12].

MAPK4 is an atypical MAPK that has not been well studied[13–15]. We recently reported that MAPK4 is a key onco-genic kinase promoting cancer via non-canonical activation of AKT/mTOR independent of PI3K/PDK1[16]. Here we report that MAPK4 is highly expressed in a large subset of TNBC. MAPK4 overexpression is sufficient to drive oncogenic outcomes, while repressing MAPK4 in the MAPK4-high human TNBC cells greatly inhibits AKT activation, cell proliferation, and anchorage-independent growth in vitro, and xenograft growth in vivo. The ability of MAPK4 to directly activate AKT suggests that elevated MAPK4 expression should reduce tumor sensitivity to PI3K blockade. In accord with this, knockdown/knockout of MAPK4 in the MAPK4-high TNBC cells and xenografts sensitized them to PI3K inhibition. These results identify MAPK4 as a promising therapeutic target for TNBC and its potential in combined ther-apy with PI3K inhibition.

## Results

**MAPK4 is highly expressed in a significant subset of human TNBC.** Analysis of 817 gene expression profiles in The Cancer Genome Atlas (TCGA)[17] revealed that MAPK4 expression is elevated in 30% or more of basal-like BCa (Fig. 1a), 70–80% of which are TNBC[18–23]. In contrast, much lower MAPK4 expres-sion was detected in the other luminal A, luminal B, HER2-amplified, and normal-like BCa types. We also analyzed MAPK4 expression in the Baylor College of Medicine BCa patient-derived xenograft (PDX) collection with completed RNA-Seq data ($n = 92$, https://pdxportal.research.bcm.edu/, public and private data combined), the majority of which are TNBC ($n = 69$, Fig. 1b, c). We also observed that MAPK4 is highly expressed in a large subset of these TNBC PDX models. Altogether, these data sup-port that MAPK4 is expressed at significant levels in a large subset of TNBC/basal-like BCa.

**MAPK4 activates AKT in human TNBC cells.** To assess MAPK4 biology in human TNBC, we first surveyed its expression in multiple commonly used TNBC lines, including HS578T, MDA-MB-231, HCC1937, SUM159, MDA-MB-468, HCC1395, and HCC1806 cells, as well as in the "normal" human mammary epithelial MCF10A cells. We observed high levels of endogenous

MAPK4 (MAPK4-high) expression in the MDA-MB-231, HS578T, and HCC1937 cells, followed by a lower MAPK4 expression in SUM159 cells (Fig. 2a). In contrast, MDA-MB-468, HCC1806, HCC1395, and the "normal" MCF10A cells express low-to-nondetectable levels of MAPK4. As an initial test of the impact of MAPK4 expression, we performed knockdown of MAPK4 (lentiviral shRNA) in the MAPK4-high MDA-MB-231, HS578T, and HCC1937 cells as well as in SUM159 cells. We observed that MAPK4 knockdown in all four TNBC cell lines repressed AKT phosphorylation and inhibited its activation, as evidenced by inhibition of GSK3β phosphorylation (Fig. 2b). We also overexpressed MAPK4 in SUM159, MDA-MB-468, HCC1395, HCC1806, and MCF10A cells in a Dox-inducible manner. In accord with loss of function results, MAPK4 over-expression induced AKT phosphorylation and activation in these cells (Fig. 2c and see below, Fig. 4d). In agreement with our previous observation in other types of human cancers[16], these results indicate that MAPK4 plays essential roles in promoting AKT phosphorylation/activation in the MAPK4-high TNBC cells.

To further confirm and extend these observations, we used CRISPR/Cas9 technology to generate *MAPK4* null BCa cell lines as previously described[16]. As expected, genetic ablation of *MAPK4* in both MDA-MB-231 and SUM159 cells markedly inhibited AKT phosphorylation and activation (Fig. 2d). Finally, we demonstrated that MAPK4 overexpression rescued AKT phosphorylation in *MAPK4*-knockout MDA-MB-231 and SUM159 cells (Fig. 2e). Altogether, these data support a critical role of MAPK4 in promoting AKT phosphorylation/activation in human TNBC cells.

**MAPK4 promotes TNBC cell growth in vitro.** We next deter-mined the impact of MAPK4 on TNBC cell growth. MAPK4 knockdown in the MAPK4-high human TNBC HCC1937 and HS578T cells, as well as in SUM159 cells, greatly inhibited their growth, including anchorage-independent growth in vitro (Fig. 3a, c–e). Accordingly, overexpression of MAPK4 promoted SUM159, HCC1395, and HCC1806 cell growth, as well as the anchorage-independent growth of SUM159 cells (Fig. 3b, g). Increased BrdU incorporation confirmed the enhanced pro-liferation of SUM159 cells in response to Dox-induced MAPK4 overexpression (Fig. 3h). These data support a crucial role of MAPK4 in promoting TNBC cell growth.

Human TNBC MDA-MB-231 cells have properties of cancer progenitor cell populations[24]. Accordingly, while MAPK4 knock-down in MDA-MB-231 cells did not significantly affect their proliferation (Supplementary Fig. 1a), it greatly repressed their anchorage-independent growth (Fig. 3f) and suppressed mammo-sphere formation (Fig. 3i). These results further support MAPK4 tumor-promoting activity in TNBC and shed light on a potential role of MAPK4 in regulating TNBC progenitor cell biology.

**MAPK4 promotes TNBC xenograft growth in vivo.** To define how MAPK4 regulates TNBC in vivo, we compared TNBC xenograft growth in severe combined immunodeficient (SCID) mice using engineered MDA-MB-231 and HCC1937 cells with Dox-inducible knockdown of MAPK4 or non-targeting control. MAPK4 knockdown robustly inhibited TNBC xenograft growth (Fig. 4a, b), while MAPK4 overexpression promoted SUM159 xenograft growth (Fig. 4c). These data further confirm the tumor-promoting activities of MAPK4 in TNBC.

We have previously shown that MAPK4 overexpression transformed the "normal" prostate epithelial PNT1A cells into anchorage-independent growth in vitro[16]. To further assess the oncogenic activity of MAPK4 in mammary epithelial cells, we investigated MAPK4 activity in transforming "normal" human

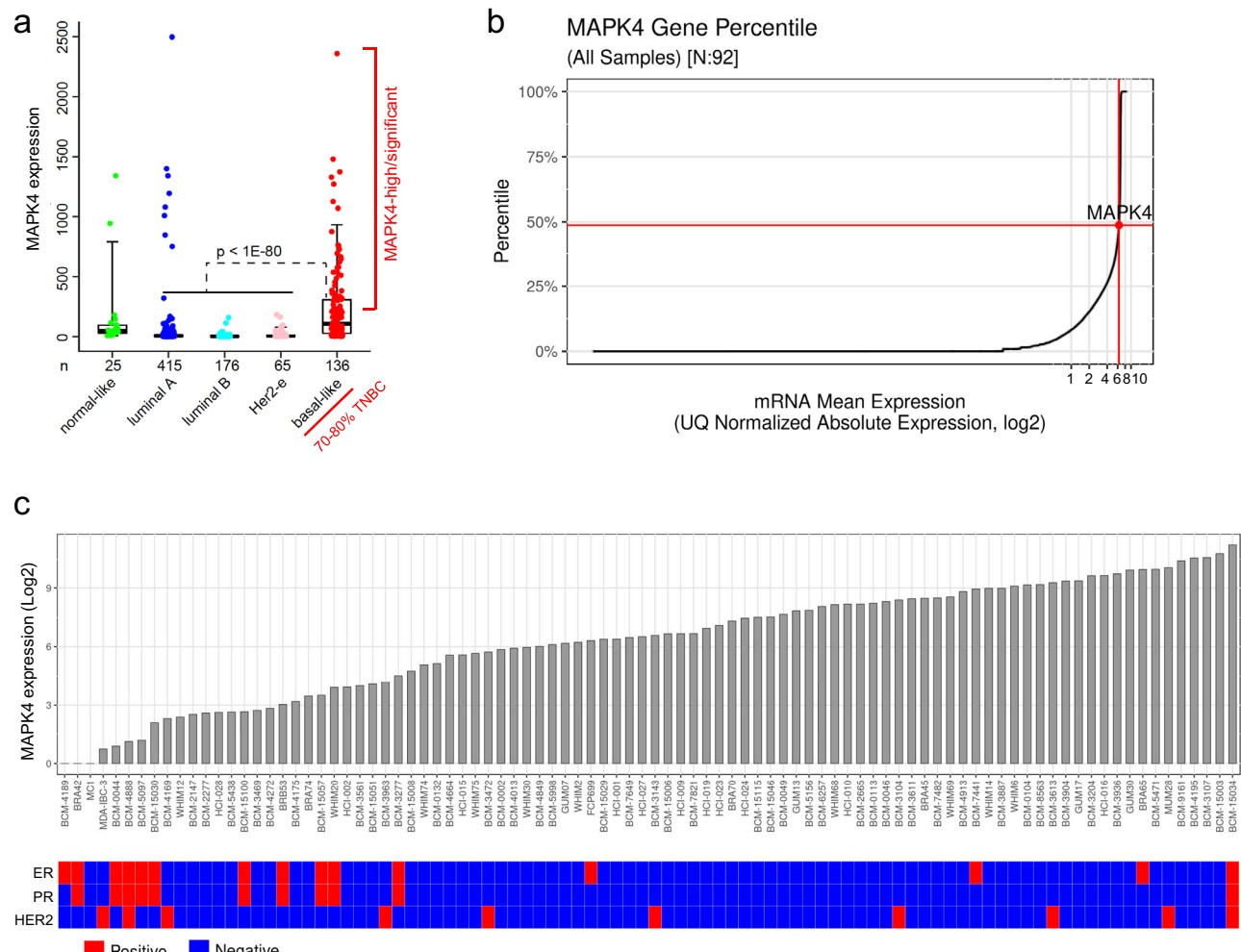

**Fig. 1 MAPK4 is highly expressed in a subset of basal-like BCa and TNBC. a** MAPK4 mRNA expression across 817 BCa from The Cancer Genome Atlas (TCGA). Boxplot represents 5% (lower whisker), 25% (lower box), 50% (median), 75% (upper box), and 95% (upper whisker). *P* value by two-sided *t* test on log2-transformed expression values. *n* represents independent patients. **b**, **c** MAPK4 mRNA expression across 92 BCa PDX models, including 69 TNBC PDX models. MAPK4 is markedly expressed (at around the 50th percentile of all genes expressed) in these PDX tumors (**b**). IHC was used for evaluating ER, PR, HER2 expression status within these tumors.

mammary epithelial MCF10A cells. As expected, MAPK4 overexpression similarly activated AKT and transformed MCF10A cells into anchorage-independent growth, further supporting the oncogenic activity of MAPK4 in mammary epithelial cells (Fig. 4d, e).

**MAPK4 promotes insulin and EGF-induced AKT activation in a parallel action of PI3K.** Due to the lack of conserved T-x-Y motif (S-E-G for MAPK4), there are no identified MAPK kinases (MAPKKs), the dual Ser/Thr and Tyr kinases, to phosphorylate and activate MAPK4. Type I P21 activated kinases 1, 2, and 3 (PAK1/2/3) have been shown to phosphorylate S186 (S189 on the closely related MAPK6) and activate the MAPK4/MAPK6-MK5 signaling cascade[25]. However, a previous study revealed no correlation between cellular MAPK4/MAPK6 phosphorylation and the extracellular stimuli or stress conditions examined[26], and the impact of this pathway on MAPK4-AKT signaling remains unknown. To determine whether extracellular stimuli can activate the MAPK4-AKT signaling cascade, we first examined AKT phosphorylation in insulin (100 nM) stimulated serum-starved MDA-MB-231 and HS578T cells with Dox-induced knockdown of MAPK4 (ishMAPK4) or control (iNT). Insulin treatment of

the control TNBC cell lines greatly stimulated AKT phosphorylation, which lasted at least 2 h. In contrast, knockdown of MAPK4 significantly reduced insulin-stimulated AKT phosphorylation at 10 min after treatment, and rapidly decreased such phosphorylation to basal levels within 1–2 h (Fig. 5a).

To investigate how PI3K pathway inhibition affects MAPK4-AKT signaling, we examined AKT phosphorylation/activation status in insulin (100 nM) stimulated serum-starved SUM159 cells with Dox-induced knockdown of MAPK4 or control. These cells were also pre-treated with PI3K inhibitors Pictilisib (20 nM), Alpelisib (100 nM), or DMSO vehicle control. In accord with the prediction of parallel pathways for AKT activation, either MAPK4 knockdown or PI3K inhibition (Pictilisib or Alpelisib) alone exhibited partial effects, while concurrent MAPK4 knockdown and PI3K inhibitor treatment robustly blocked insulin-induced AKT phosphorylation/activation (Fig. 5b).

To further expand these observations, we performed a similar study in EGF (200 ng/ml) stimulated serum-starved MDA-MB-231 and SUM159 cells with Dox-induced knockdown of MAPK4 or control. These cells were also pre-treated with PI3K inhibitors Pictilisib (20 nM), Alpelisib (100 nM), or DMSO vehicle control. Similar to insulin treatment, EGF treatment of the control TNBC cell lines greatly promoted AKT phosphorylation and activation.

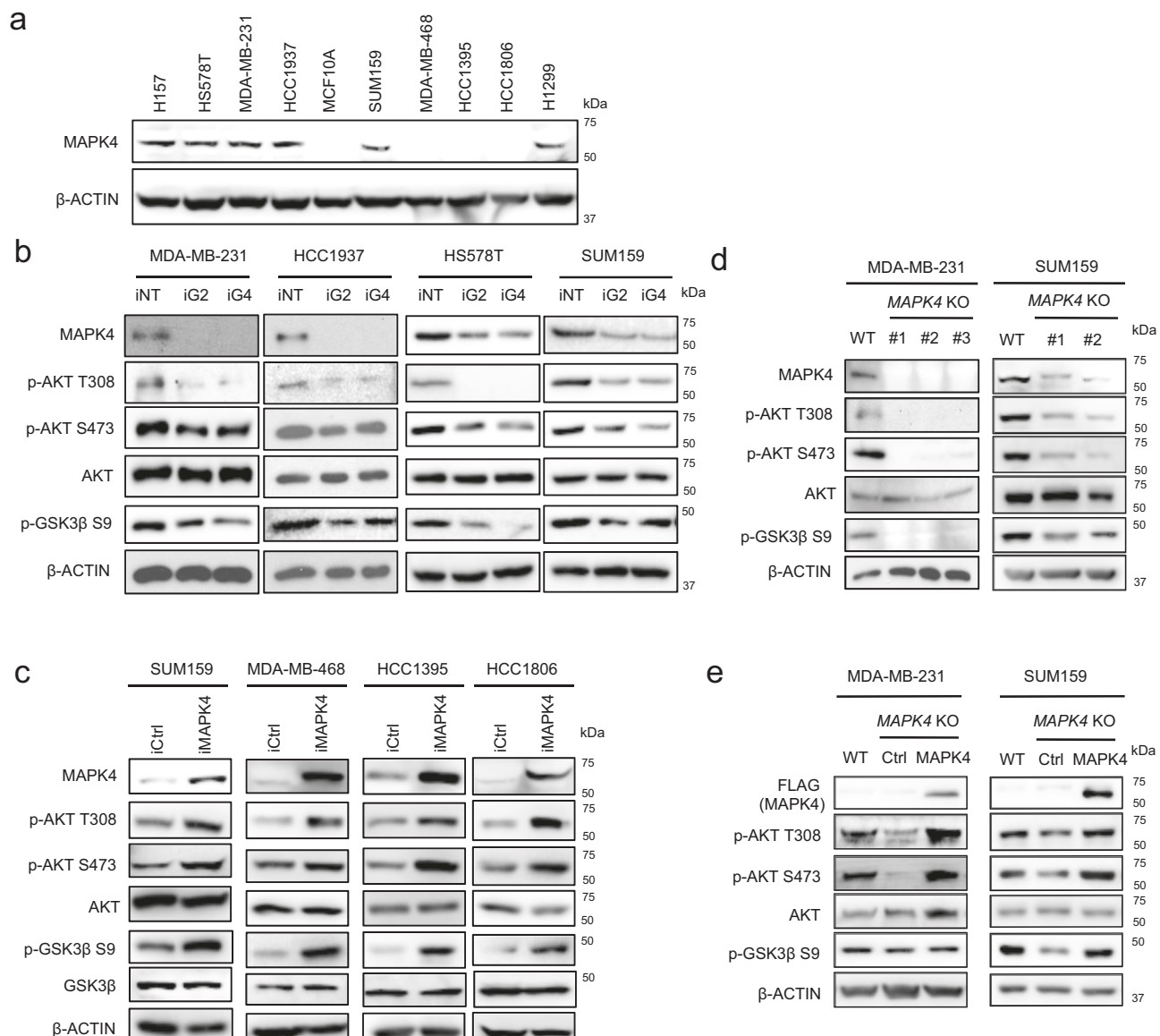

**Fig. 2 MAPK4 activates AKT in human TNBC cells. a** Western blots on MAPK4 expression in various human TNBC cell lines and MCF10A, a "normal" human mammary epithelial cell line. H157 and H1299 are human non-small cell lung cancer cell lines expressing high levels of MAPK4 as we previously reported. **b** Western blots on engineered MDA-MB-231, HCC1937, HS578T, and SUM159 cells with 4 μg/ml Dox-induced knockdown of MAPK4 (iG2, iG4) or control (iNT). **c** Western blots on engineered SUM159, MDA-MB-468, HCC1395, and HCC1806 cells with 0.5 μg/ml Dox-induced expression of MAPK4 (iMAPK4) or control (iCtrl). **d** CRISPR/Cas9 technology was used to knockout *MAPK4* in MDA-MB-231 cells (clone# 1, 2, 3) and SUM159 cells (clone# 1, 2). Western blots were used to compare AKT phosphorylation and activation among these cells. **e** MAPK4 was ectopically expressed in the MDA-MB-231 *MAPK4*-knockout cells (clone# 3) and SUM159 *MAPK4*-knockout cells (clone# 2). Western blots were used to detect AKT phosphorylation and activation. Data are representative of at least three independent experiments.

Again, while either knockdown of MAPK4 or PI3K inhibitor (Pictilisib or Alpelisib) treatment alone exhibited partial effect, concurrent knockdown of MAPK4 and PI3K inhibitor treatment robustly blocked EGF-induced AKT phosphorylation/activation (Fig. 5c). Altogether, these data strongly support that extracellular stimuli including insulin and EGF can activate the MAPK4-AKT signaling axis in a pathway that parallels the action of PI3K.

**AKT activation is critical for MAPK4 tumor-promoting activity in TNBC.** We previously demonstrated that AKT activation is critical for the tumor-promoting activity of MAPK4[16,27]. To further confirm this in TNBC, we examined how the AKT inhibitors MK2206, GSK2141795, and/or GDC-0068 affect growth of the engineered HCC1395, HCC1806, and SUM159

cells with 0.5 μg/ml Dox-induced overexpression of MAPK4 (iMAPK4) or control (iCtrl). These AKT inhibitor treatments largely abolished the MAPK4 activities in enhancing the growth of HCC1395, HCC1806, and SUM159 cells as well as MAPK4 activities in promoting the anchorage-independent growth of SUM159 cells (Fig. 6a–d). These data confirm that, as we showed in other cancer types[16,27], AKT is also a key node for mediating MAPK4 activity in promoting TNBC.

**MAPK4 reduces TNBC sensitivity to PI3K inhibition.** An α-isoform-specific PI3K inhibitor Piqray (Alpelisib) was recently approved to treat hormone receptor-positive, HER2-negative, PIK3CA-mutated, advanced or metastatic BCa after progression on endocrine therapy. However, PI3K inhibitors have shown

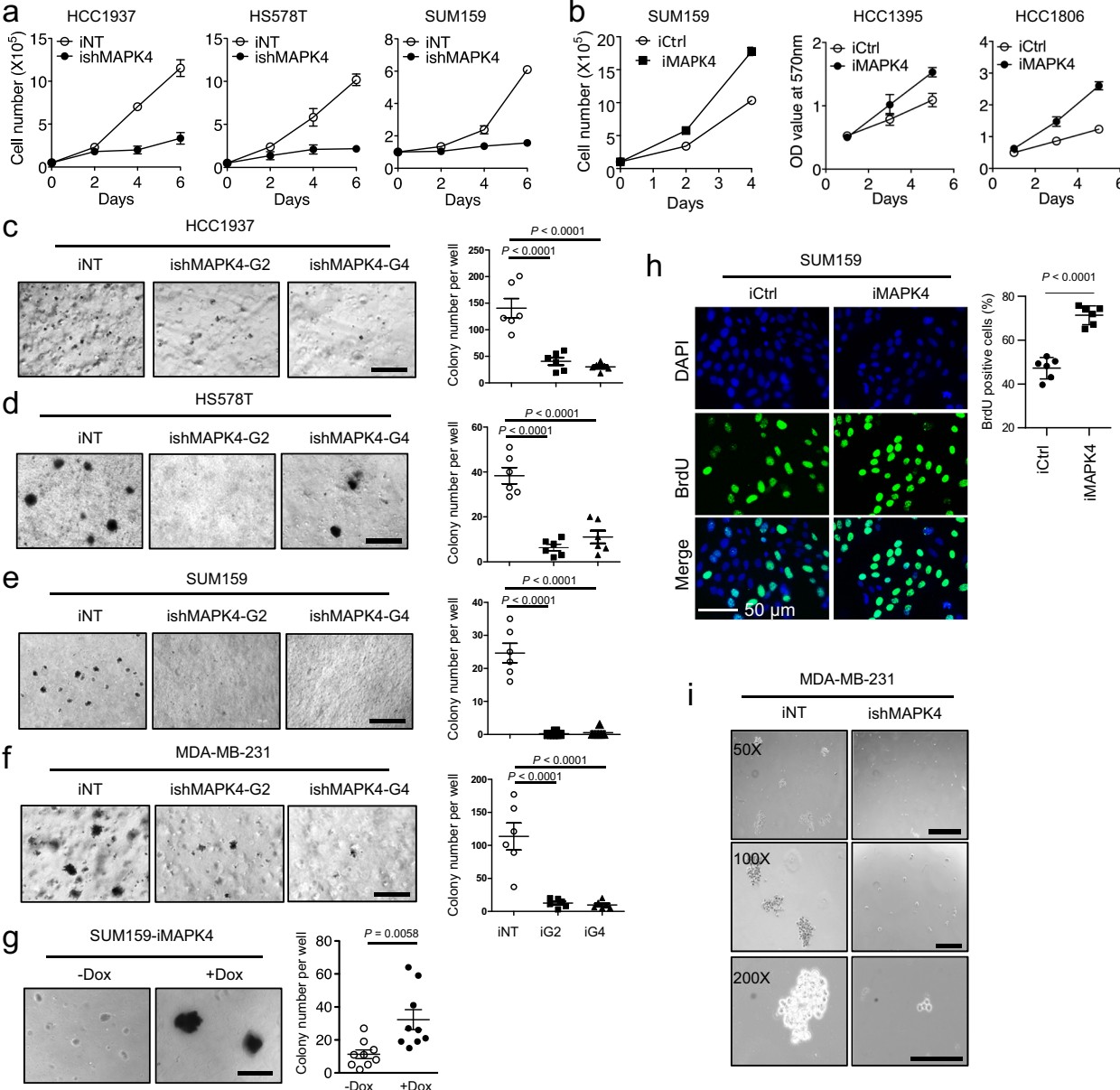

**Fig. 3 MAPK4 promotes TNBC cell growth in vitro.** Proliferation assays comparing the growth of **a** engineered HCC1937, HS578T, and SUM159 cells with 4 μg/ml Dox-induced knockdown of MAPK4 (ishMAPK4) or control (iNT), and **b** SUM159, HCC1395, and HCC1806 cells with 0.5 μg/ml Dox-induced expression of MAPK4 (iMAPK4) or control (iCtrl). Soft-agar assays comparing the anchorage-independent growth of the engineered **c** HCC1937, **d** HS578T, **e** SUM159, **f** MDA-MB-231 cells with 4 μg/ml Dox-induced knockdown of MAPK4 (ishMAPK4-G2 or G4) or control (iNT), and **g** SUM159 cells with 0 or 0.5 μg/ml Dox-induced expression of MAPK4 (iMAPK4). Bar: 500 μm. **h** BrdU incorporation assays on the engineered SUM159 cells with 0.5 μg/ml Dox-induced ectopic expression of MAPK4 (iMAPK4) or control (iCtrl). Right panels in **c–h** show data quantification. **i** Mammosphere assay on MDA-MB-231 cells with 4 μg/ml Dox-induced knockdown of MAPK4 (ishMAPK4) or control (iNT). Bar: 500 μm (50X), 200 μm (100X and 200X). Data are mean ± SEM (**a**, **c–g**) and mean ± SD (**b**, **h**). *P* values determined by unpaired two-tailed Student's *t* test and adjusted *P* values determined by one-way ANOVA followed by Dunnett's multiple comparisons. Data are representative of at least three independent experiments. Source data are provided as a Source data file.

limited therapeutic effects in TNBC[11,12]. The apparent parallel mode of MAPK4 and PI3K actions in activating AKT in TNBC (Fig. 5) would predict that repressing MAPK4 should sensitize MAPK4-high TNBC cells to PI3K blockade.

To define how MAPK4 affects TNBC cell response to PI3K blockade, we performed colony formation assays on the engineered HCC1937, HS578T, MDA-MB-231, and SUM159 cells with 4 μg/ml Dox-induced knockdown of MAPK4 (iG2 and iG4) or control (iNT). We treated these cells with increasing doses of PI3K inhibitors Pictilisib, Alpelisib, or DMSO vehicle control.

Knockdown of MAPK4 both significantly repressed MDA-MB-231, SUM159, HS578T, and HCC1937 cell growth and sensitized them to both Pictilisib and Alpelisib treatments (Fig. 7a–d; Supplementary Fig. 1b). We further confirmed that knockdown of MAPK4 also sensitized SUM159 cells to another commonly used PI3K inhibitor LY294002 (2 μM), and the effects were comparable to treatments using Pictilisib (1 μM) and Alpelisib (0.5 μM, Fig. 7e). Interestingly, after 10 days of culture in the clonogenic assay settings (cells plated at low density for clonal growth of individual cells), the MAPK4-knockdown SUM159 and

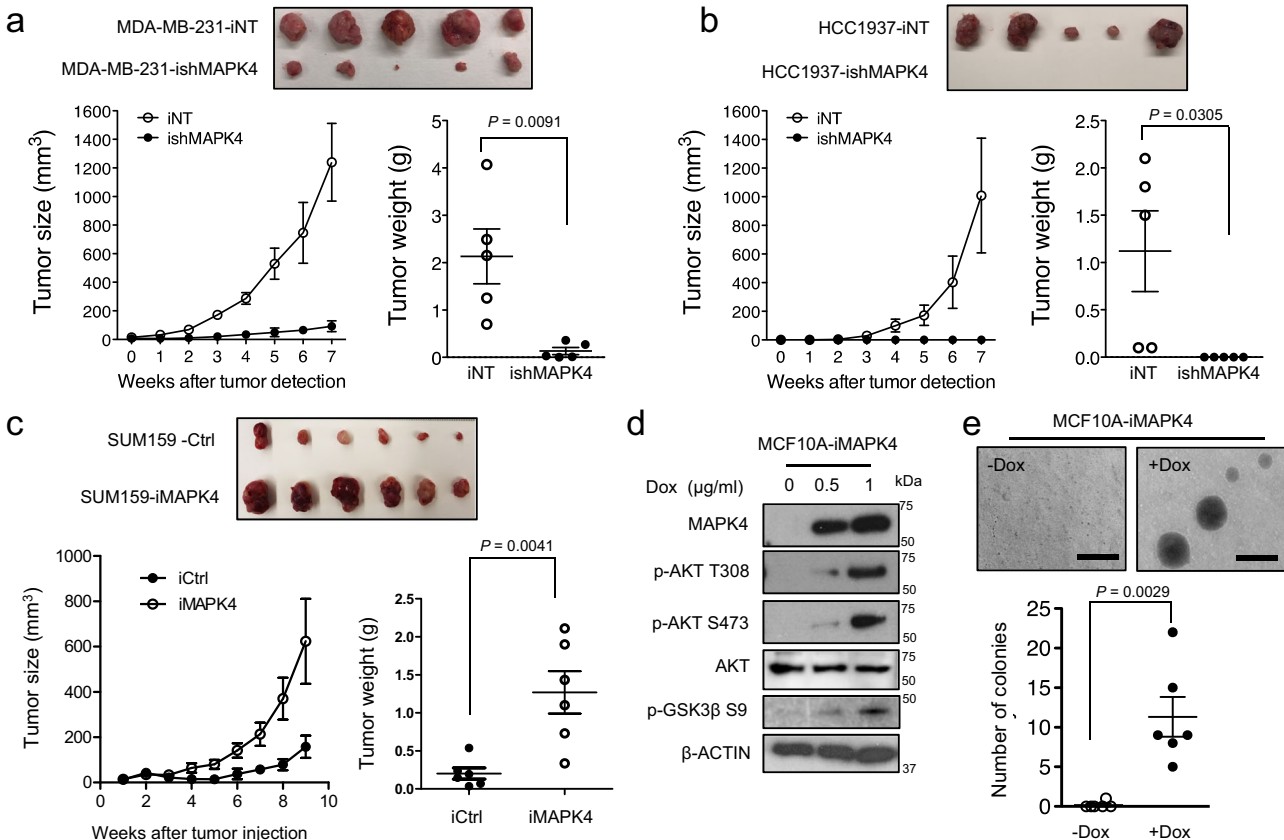

**Fig. 4 MAPK4 promotes TNBC xenograft growth in vivo.** Dox-induced knockdown of MAPK4 inhibits **a** MDA-MB-231 and **b** HCC1937 xenograft growth in SCID mice. **c** Dox-induced overexpression of MAPK4 promotes SUM159 xenograft growth in SCID mice. $2 \times 10^6$ of engineered MDA-MB-231 or HCC1937 cells with Dox-inducible knockdown of MAPK4 (ishMAPK4) or control (iNT), or engineered SUM159 with Dox-inducible ectopic expression of MAPK4 (iMAPK4) or control (iCtrl), were injected into the mammary fat pad of female SCID mice (iCtrl or iNT on the left side; iMAPK4 or ishMAPK4 on the right side). Mice began receiving Dox (0.2 mg/ml for SUM159 xenografts and 4 mg/ml for MDA-MB-231 and HCC1937 xenografts) in 1–10% sucrose in drinking water on the day of xenograft implantation. Tumors were harvested as indicated. **d** Western blots and **e** Soft-agar assays on engineered MCF10A cells with Dox-inducible MAPK4 expression. Dox concentration used are as indicated in (**d**) and 0.5 μg/ml in (**e**). Bar: 500 μm. Data are mean ± SEM. *P* values determined by unpaired two-tailed Student's *t* test. Data are representative of at least three independent experiments. Source data are provided as a Source data file.

MDA-MB-231 cells partially regained AKT phosphorylation/ activation, presumably due to PI3K-AKT pathway activation for individual cell growth/survival. In accord with this, AKT phosphorylation/activation in these MAPK4-knockdown cells was highly sensitive to PI3K blockade (0.5 μM Alpelisib), further confirming MAPK4-AKT as the essential pathway for regulating TNBC sensitivity to PI3K inhibition (Fig. 7f).

To define whether MAPK4 overexpression makes TNBC cells less sensitive to PI3K blockade, we also performed colony formation assays on the engineered SUM159, MDA-MB-468, HCC1395, and HCC1806 cells with 0.5 μg/ml Dox-induced ectopic expression of MAPK4 (iMAPK4) or control (iCtrl). We similarly treated these cells with increasing doses of PI3K inhibitors Pictilisib, Alpelisib, or DMSO vehicle control. As expected, MAPK4 overexpression both significantly promoted SUM159, MDA-MB-468, HCC1395, and HCC1806 cell growth and reduced their sensitivity to both Pictilisib and Alpelisib (Fig. 8a–d; Supplementary Fig. 1c). Accordingly, the AKT phosphorylation/activation was markedly maintained in the MAPK4-overexpressing SUM159 and HCC1806 cells under PI3K inhibitor treatments (compared to similarly treated iCtrl cells), further confirming MAPK4-AKT as the essential pathway for regulating TNBC sensitivity to PI3K inhibition (Fig. 8e).

In agreement with the colony formation assay results, knockdown of MAPK4 both significantly repressed the anchorage-independent

growth of the MAPK4-high HCC1937 and MDA-MB-231 cells and greatly sensitized them to PI3K inhibitors LY294002 (5 μM), Alpelisib (1 μM), and Pictilisib (1 μM, Fig. 9a, b). SUM159 cells express lower levels of MAPK4 (Fig. 2a), yet knockout of *MAPK4* from SUM159 cells both inhibited their anchorage-independent growth and sensitized them to PI3K inhibitors LY294002 (2 μM), Pictilisib (0.5 μM), and Alpelisib (0.5 μM, Fig. 9c). In contrast, ectopic expression of MAPK4 in these *MAPK4*-KO SUM159 cells largely rescued their growth and reduced their sensitivity to PI3K inhibition (Fig. 9c). In accord with these results, knockdown of MAPK4 in the MAPK4-high MDA-MB-231 cells inhibited and overexpression of MAPK4 in the MAPK4-low HCC1806 cells promoted their anchorage-independent growth and increased MDA-MB-231 and reduced HCC1806 cell sensitivity to Pictilisib (Fig. 9d, e).

Altogether, these results support a crucial role of MAPK4 in not only driving TNBC cell growth but also reducing their sensitivity to PI3K inhibition.

**MAPK4-knockout TNBC xenografts are sensitive to PI3K inhibitor Alpelisib.** To further assess whether MAPK4 blockage can sensitize TNBC tumors to PI3K inhibition, we examined how *MAPK4* knockout affects MDA-MB-231 cell and xenograft growth and sensitivity to PI3K blockade. Consistent with the MAPK4-knockdown data (Fig. 3f and Fig. 9b, d), *MAPK4*-KO

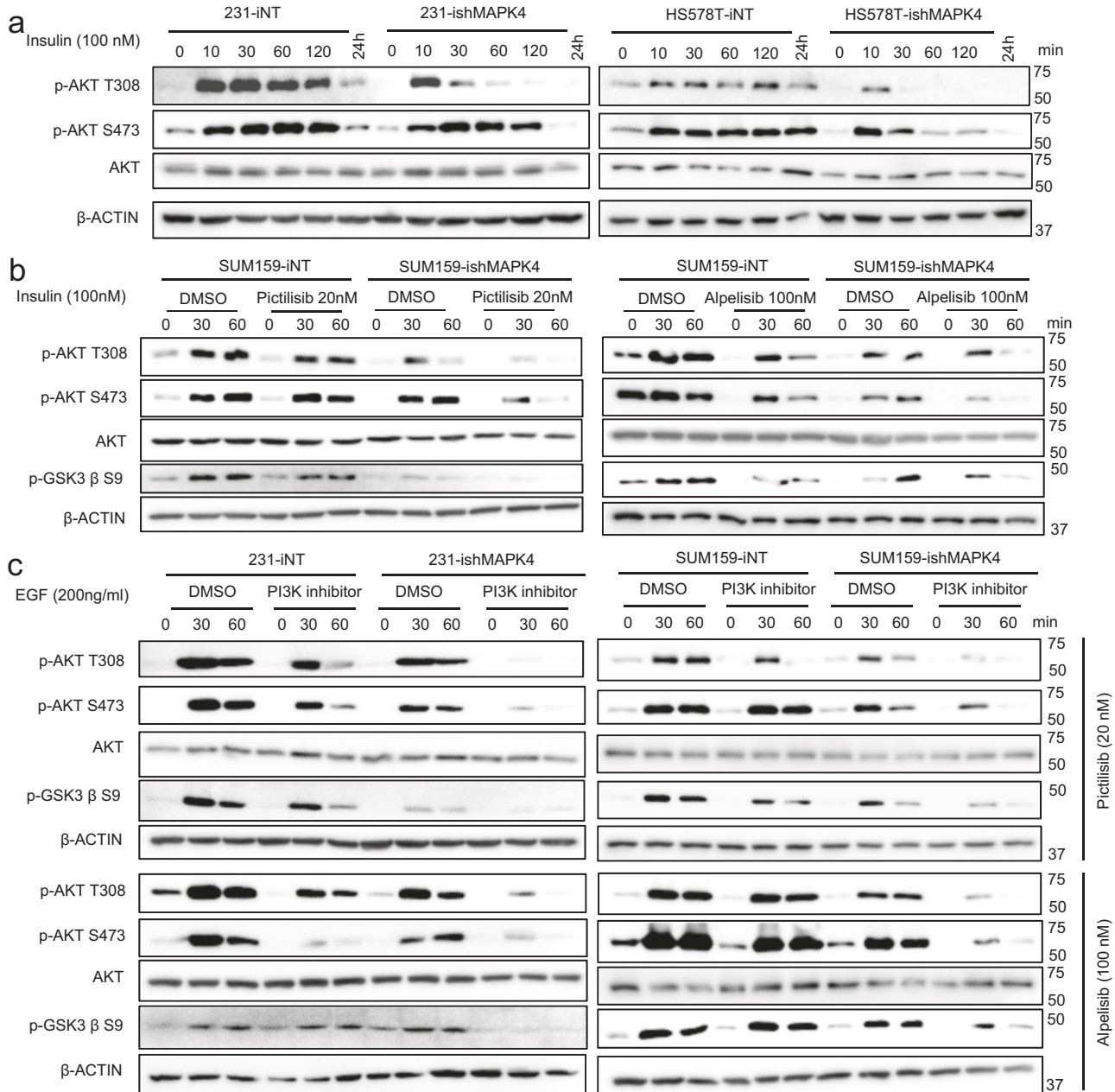

**Fig. 5 MAPK4 enhances insulin and EGF-induced AKT phosphorylation/activation independent of PI3K. a** MDA-MB-231 and HS578T cells with Dox-inducible knockdown of MAPK4 (ishMAPK4) or control (iNT) were induced with 4 μg/ml Dox for 3 days. The cells were then plated in 6-well plates. Twenty hours later, the cells were serum-starved overnight followed by treatments with 100 nM insulin for the indicated time (minutes). The cell lysates were then prepared and used in western blots. **b** Western blots on SUM159-ishMAPK4 and -iNT cells that were similarly treated as described above. These cells were also pre-treated with Pictilisib or DMSO control for 2 h before 100 nM insulin stimulation. **c** Western blots on MDA-MB-231 and SUM159 cells with Dox-inducible knockdown of MAPK4 (ishMAPK4) or control (iNT) that were similarly treated as described above. These cells were pre-treated with PI3K inhibitors Pictilisib, Alpelisib, or control for 2 h followed by 200 ng/ml EGF stimulation for the indicated time. Data are representative of at least three independent experiments.

MDA-MB-231 cells formed fewer colonies and were more sensitive to PI3K inhibitors LY294002, Pictilisib, and Alpelisib in the soft-agar assays (Fig. 10a). Ectopic MAPK4 expression both promoted the anchorage-independent growth of these *MAPK4*-KO cells and largely rescued their growth in the presence of PI3K inhibitors (Fig. 10a).

We next performed xenograft studies in SCID mice using parental and *MAPK4*-KO MDA-MB-231 cells. Consistent with our above data from the MAPK4-knockdown MDA-MB-231 xenografts (Fig. 4a), knockout of *MAPK4* also significantly repressed MDA-MB-231 xenograft growth (Fig. 10b, c). The PI3K inhibitor

Alpelisib showed minimal effect on the growth of the parental MDA-MB-231 xenografts. In contrast, Alpelisib significantly inhibited the growth of *MAPK4*-KO MDA-MB-231 xenografts (Fig. 10b, c), supporting that loss of MAPK4 sensitizes TNBC tumors to PI3K inhibition in vivo.

## Discussion

Profiling human tumors show that MAPK4 is highly expressed in a large subset of TNBC. We estimate that this includes 30% or more of basal-like BCa/TNBC, with lesser amounts in other

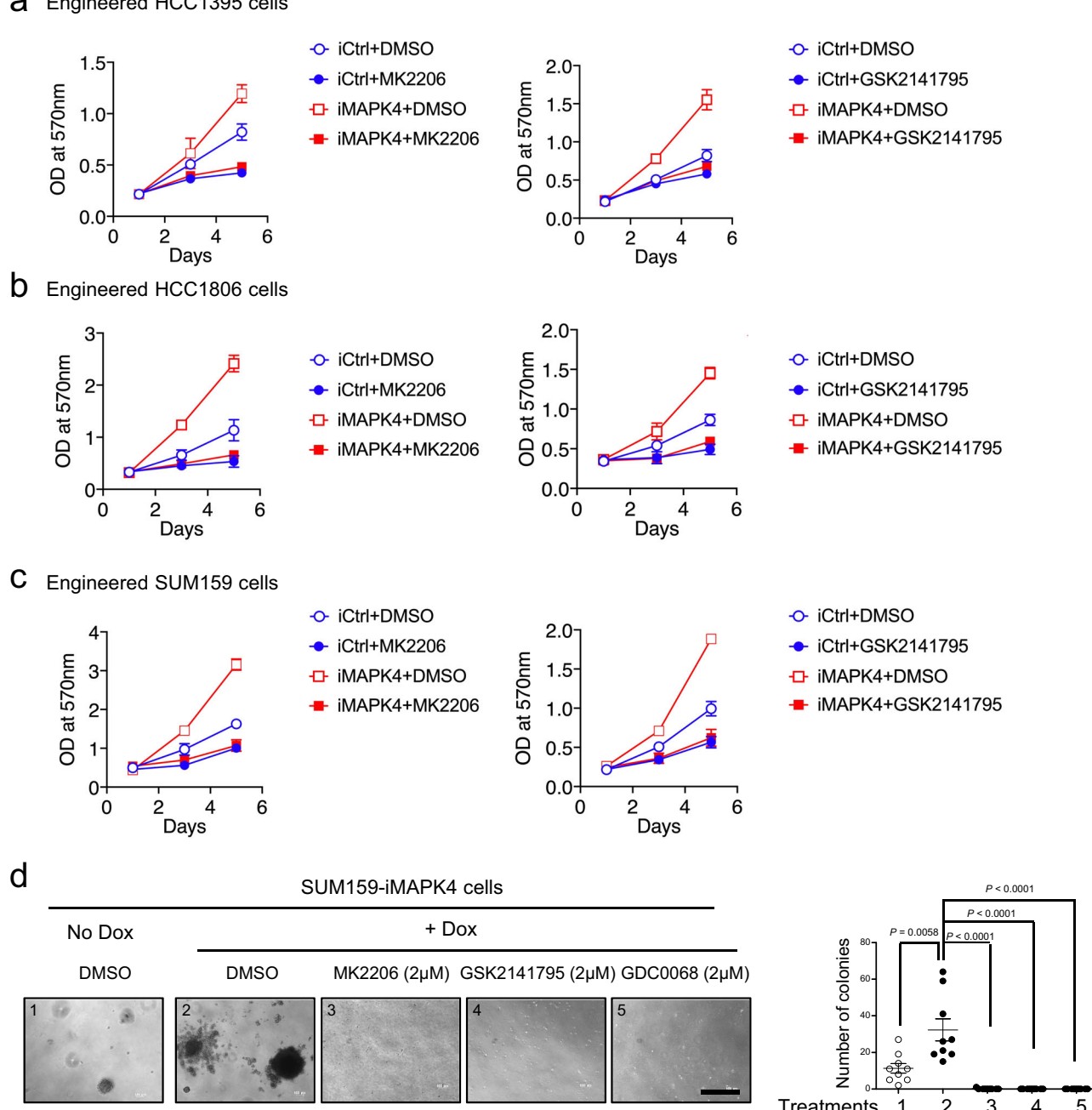

**Fig. 6 AKT inhibition blocks MAPK4 tumor-promoting activity in TNBC.** Proliferation assays on the engineered **a** HCC1395, **b** HCC1806, and **c** SUM159 cells with 0.5 μg/ml Dox-induced expression of MAPK4 (iMAPK4) or control (iCtrl). The cells were also treated with 1 μM of the AKT inhibitors MK2206 (left) or GSK2141795 (right) or control. Data are mean ± SD. **d** Representative images and quantification of soft-agar assays on Dox-induced SUM159-iMAPK4 cells treated with AKT inhibitors MK2206, GSK2141795, GDC-0068, and DMSO control. Also shown are the control-treated SUM159-iMAPK4 cells without Dox induction. Bar: 500 μm. Data are mean ± SEM. *P* value by unpaired two-tailed Student's *t* test. Data are representative of at least 3 independent experiments. Source data are provided as a Source data file.

subtypes. In accord with our earlier results in other cancers[16,27], MAPK4 knockdown or knockout in these MAPK4-high TNBC cell lines greatly repressed AKT activation, cell growth, and xenograft growth. In contrast, MAPK4 promoted cell migration in the wound healing assay in only one out of four TNBC cell lines tested (Supplementary Fig. 2). Therefore, unlike the growth-promoting activity, MAPK4's ability to promote cell motility appears cell-context dependent.

Our previous results also showed that MAPK4 expression can transform the normal prostate epithelial PNT1A cells into both anchorage-independent growth in vitro[16] and orthotopic xenograft tumor growth in vivo (unpublished observation). Our current results show that MAPK4 can transform normal human mammary epithelial MCF10A cells into anchorage-independent growth in vitro but failed to transform them into tumor growth in vivo six months after initial inoculation into mammary fat pad (Fig. 4e and unpublished observation). Overall, these data define MAPK4 as a bona fide oncogene and indicate that the MAPK4-high subset may represent a unique subtype of TNBC that would be sensitive to MAPK4 inhibition. Interestingly, five out of the

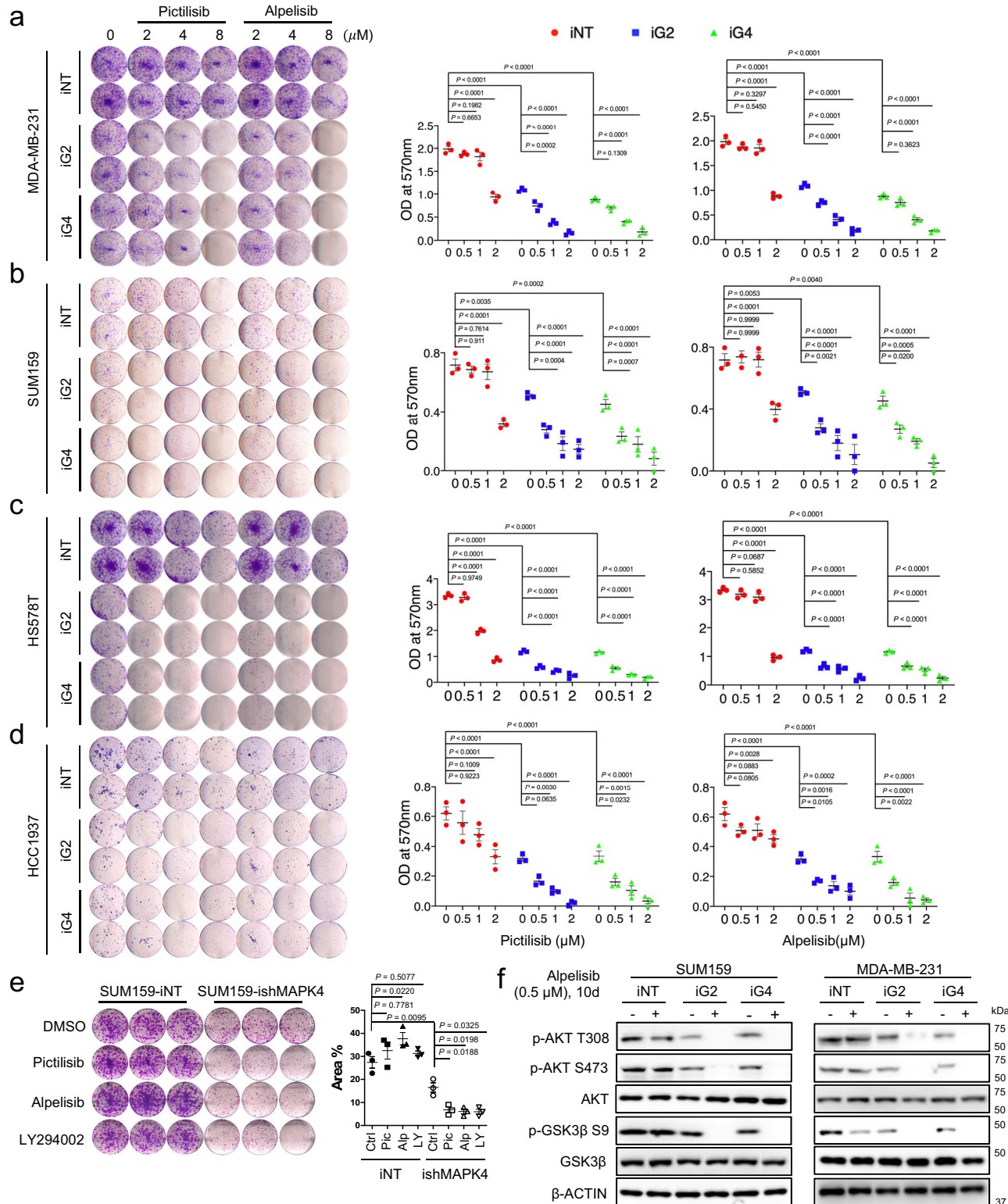

**Fig. 7 Knockdown of MAPK4 sensitizes human TNBC cells to PI3K inhibitor treatments.** Representative images and quantification of colony formation assay data of engineered **a** MDA-MB-231, **b** SUM159, **c** HS578T, and **d** HCC1937 cells with 4 µg/ml Dox-induced knockdown of MAPK4 (iG2, iG4) or control (iNT). The cells were also treated with increasing doses of Pictilisib, Alpelisib, or control. **e** Representative images and quantification of SUM159-iNT and -ishMAPK4 cells in the colony formation assays. The cells were treated for 10 days with PI3K inhibitors Pictilisib (1 µM), Alpelisib (0.5 µM), LY294002 (2 µM), or DMSO. The right panels in **a–e** show quantification of colonies formed under each treatment condition described in the left panels. Data are mean ± SD (**a–d**) and mean ± SEM (**e**). Adjusted *P* values determined by two-way ANOVA followed by Sidak's multiple comparisons. **f** Western blots on SUM159 and MDA-MB-231 cells with 4 µg/ml Dox-induced knockdown of MAPK4 (iG2, iG4) or control (iNT) cells after 10 days culturing in the presence of Alpelisib (+) or DMSO (−). Data are representative of at least three independent experiments. Source data are provided as a Source data file.

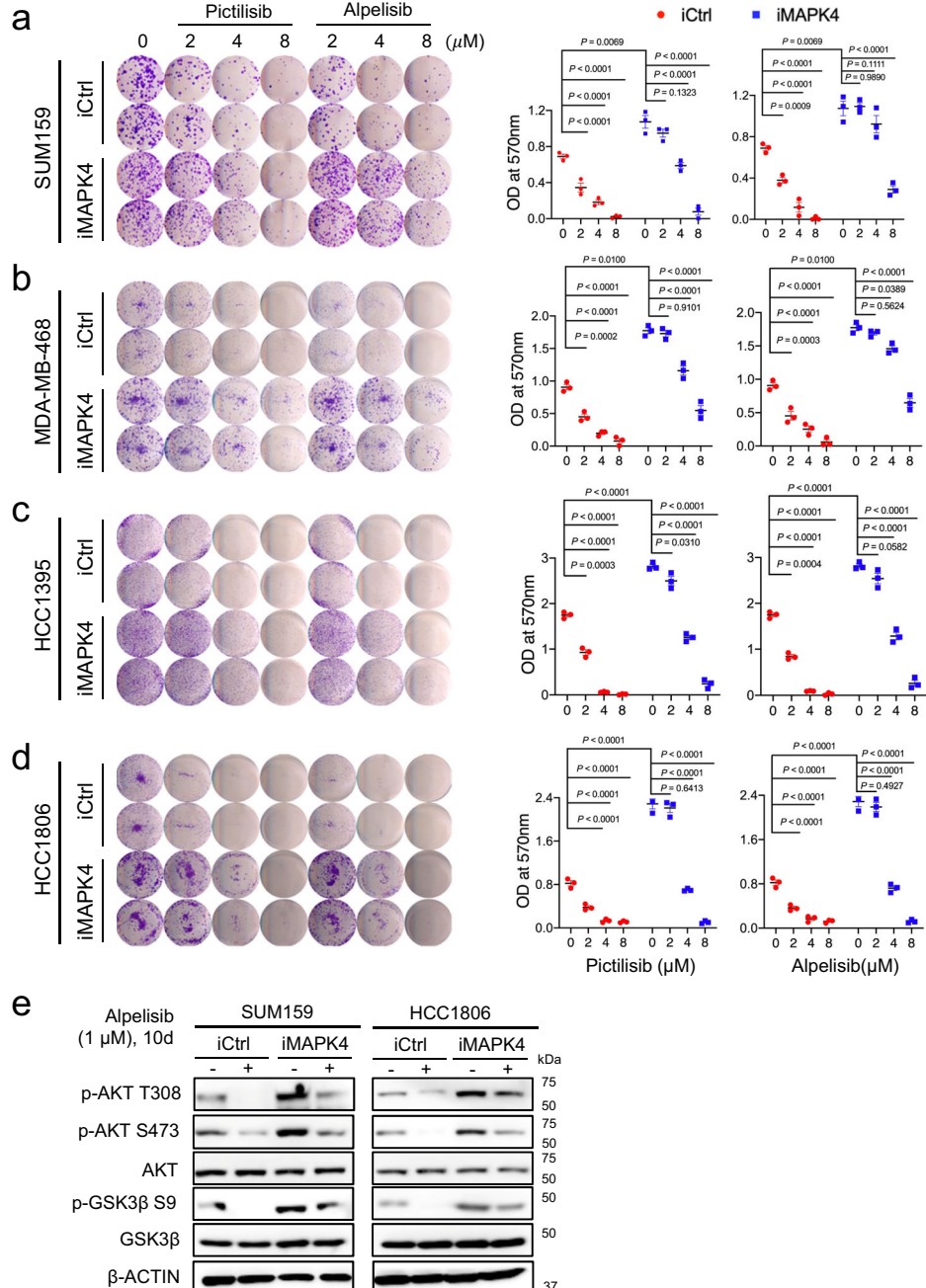

**Fig. 8 Ectopic expression of MAPK4 reduces human TNBC cell sensitivity to PI3K inhibitor treatments.** Representative images and quantification of colony formation assay data of engineered **a** SUM159, **b** MDA-MB-468, **c** HCC1395, and **d** HCC1806 cells with 0.5 µg/ml Dox-induced ectopic expression of MAPK4 (iMAPK4) or control (iCtrl). The cells were also treated with increasing doses of Pictilisib, Alpelisib, or control. The right panels show quantification of colonies formed under each treatment condition described in the left panels. Data are mean ± SD. Adjusted P values determined by two-way ANOVA followed by Sidak's multiple comparisons. **e** Western blots on SUM159 and HCC1806 cells with 0.5 µg/ml Dox-induced ectopic expression of MAPK4 (iMAPK4) or control (iCtrl) cells after 10 days culturing in the presence of Alpelisib (+) or DMSO (−). Data are representative of at least three independent experiments. Source data are provided as a Source data file.

seven TNBC cell lines that we have tested are either *PTEN*-null (HCC1395, HCC1937, and MDA-MB-468 cells with frameshift mutation or homodeletion of *PTEN*) or *PIK3CA* (SUM159) or *PIK3R1* mutated (HS578T). We are investigating the detailed molecular mechanism underlying the maintenance of the MAPK4-addiction phenotype in these TNBC cells. Although there are currently no identified specific MAPK4 inhibitors, the current results provide further impetus for their development and for future studies to critically examine their efficacy, either as

monotherapy or in combination with chemotherapy or radiation therapy, in treating MAPK4-high TNBC.

MAP kinases (MAPKs) play critical roles in mediating cell response to extracellular signals. In the canonical MAPK pathway, MAPKs are phosphorylated at both S/T and Y residues in the conserved T-x-Y motif and activated by MAPK kinases (MAPKKs), the dual Ser/Thr and Tyr kinases. In MAPK4, this T-x-Y motif is replaced by S-E-G (aa186–188), which lacks the key Y residue. Therefore, there is no identified MAPKK to phosphorylate/activate

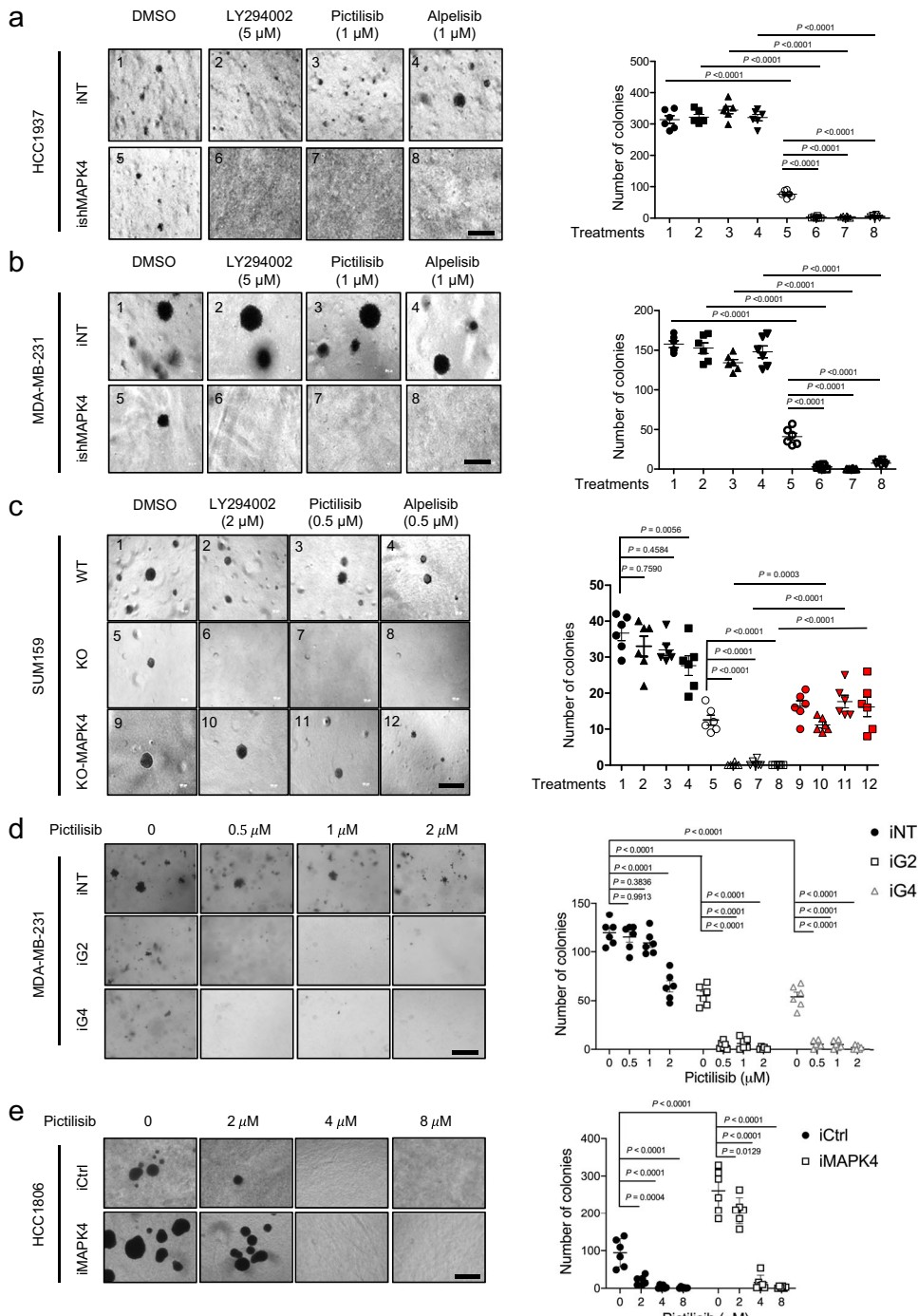

**Fig. 9 MAPK4 profoundly affects the anchorage-independent growth of TNBC cells and their response to PI3K inhibitor treatments.** Representative images and quantification of soft-agar assay data of engineered **a** Dox-induced HCC1937-iNT and -ishMAPK4, **b** Dox-induced MDA-MB-231-iNT and -ishMAPK4, and **c** wild type (WT) and *MAPK4*-knockout (KO, clone #2) SUM159 cells, as well as *MAPK4*-KO SUM159 cells with ectopic expression of MAPK4 (KO-MAPK4). The cells were also treated with PI3K inhibitors LY294002, Pictilisib, Alpelisib at the indicated concentrations, or vehicle control (DMSO). Representative images and quantification of soft-agar assay data of engineered **d** MDA-MB-231 cells with 4 μg/ml Dox-induced knockdown of MAPK4 (iG2, iG4) or control (iNT), and **e** HCC1806 cells with 0.5 μg/ml Dox-induced overexpression of MAPK4 (iMAPK4) or control (iCtrl). The cells were also treated with increasing dosages of Pictilisib at indicated concentrations. The right panels show quantification of colonies formed under each treatment condition described/numbered in the left panels. Bar: 500 μm. Data are mean ± SEM (**a**–**c**) or mean ± SD (**d**, **e**). Adjusted *P* values determined by two-way ANOVA followed by Sidak's multiple comparisons. Data are representative of at least three independent experiments. Source data are provided as a Source data file.

MAPK4. Furthermore, MAPK4 phosphorylation was not associated with extracellular stimuli or stress conditions examined[26], leading to a vague conception that MAPK4 activation is not subjected to such conditions. In contrast, our study demonstrated that the MAPK4-

AKT signaling cascade can be activated by both insulin and EGF, two key factors regulating physiology and diseases, including cancers. Further studies are needed to assess the detailed mechanism underlying how extracellular signals activate the MAPK4-AKT pathway.

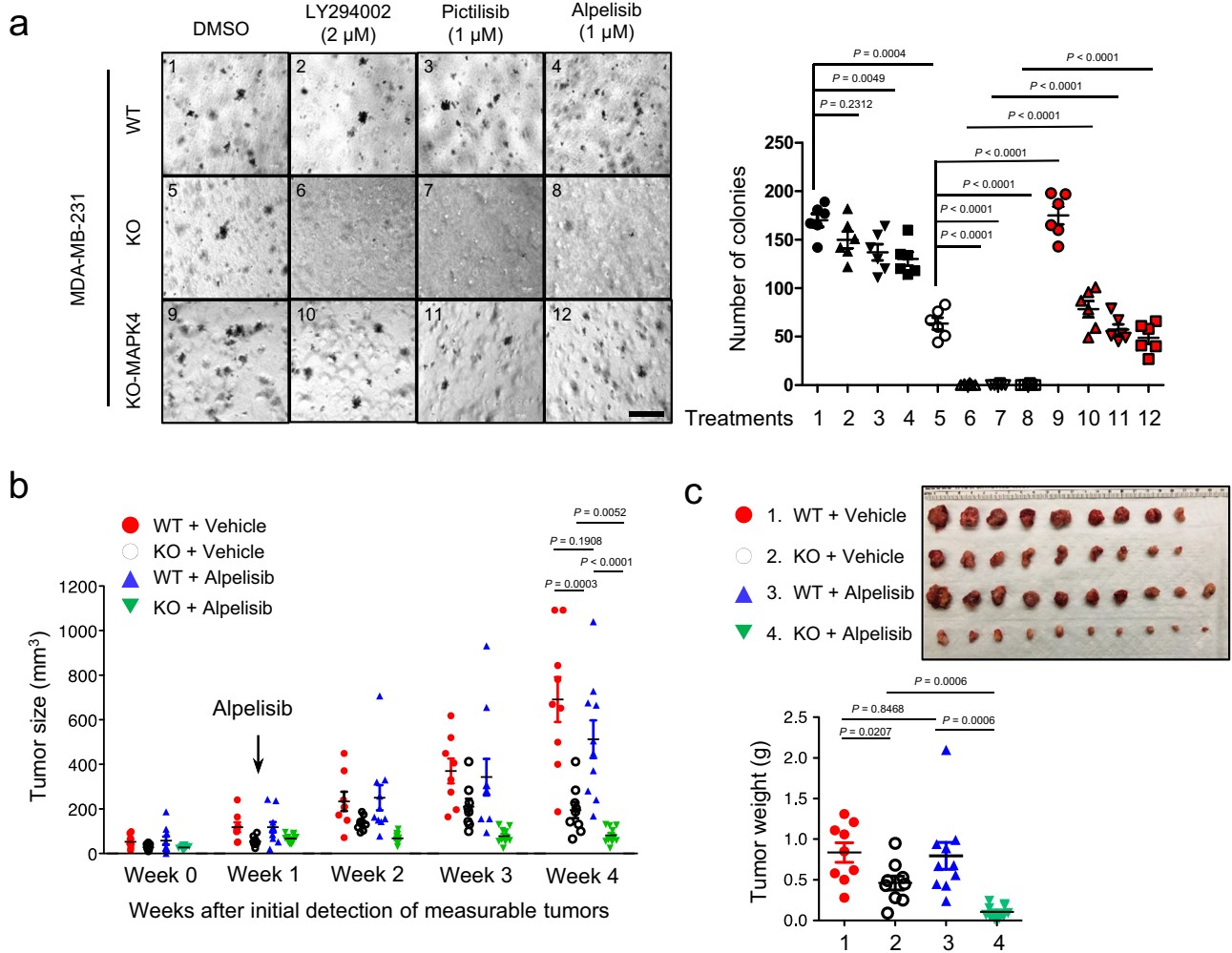

**Fig. 10 Knockout of *MAPK4* sensitizes MDA-MB-231 cells and xenografts to PI3K inhibition. a** Soft-agar assays for anchorage-independent growth of wild type (WT) and *MAPK4*-knockout (KO, clone #3) MDA-MB-231 cells, as well as *MAPK4*-KO MDA-MB-231 cells with ectopic expression of MAPK4 (KO-MAPK4), in the presence of PI3K inhibitors LY294002, Pictilisib, Alpelisib, or vehicle control (DMSO). Bar: 500 μm. The right panels show quantification (mean ± SEM) of colonies formed under each treatment condition described/numbered in the left panels. Adjusted *P* values determined by two-way ANOVA followed by Sidak's multiple comparisons. Data are representative of at least three independent experiments. **b** The weekly measurement of the growth (sizes) of wild type (WT) and *MAPK4*-knockout (KO) MDA-MB-231 xenograft tumors with 3-week continuous treatments of Alpelisib (daily oral gavage at 20 mg/kg) or Vehicle. Week 0 indicates the initial time point when measurable xenografts were detected. Arrow indicates beginning of Alpelisib treatment one week after tumor detection. **c** Xenograft tumors and tumor weights at collection. Xenograft tumor data are representative of two independent experiments. Data are mean ± SEM. *P* values determined by unpaired two-tailed Student's *t* test. Source data are provided as a Source data file.

Despite the success of the PI3K inhibitor Alpelisib in treating hormone receptor-positive, HER2-negative, PIK3CA-mutated, advanced or metastatic BCa, the therapeutic effect of PI3K inhibitors in TNBC is limited[11,12]. Both intrinsic and acquired resistance may be responsible for the lack of efficacy of PI3K inhibition in TNBC. Our identified PI3K-independent MAPK4-AKT signaling axis may provide a pathway for MAPK4-high TNBC tumor intrinsic resistance to PI3K inhibitors. Indeed, without exception, knockdown or knockout of MAPK4 in the MAPK4-high MDA-MB-231, HCC1937, HS578T, as well as MAPK4-medium SUM159 cells sensitized them to PI3K inhibition in vitro. Knockout of *MAPK4* in the MDA-MB-231 xenografts both repressed their growth and sensitized them to Alpelisib in vivo. These data support the prediction that inhibiting MAPK4 should sensitize MAPK4-high TNBC to PI3K inhibition. Future studies are needed to test this prediction using MAPK4-specific inhibitors in combination with PI3K inhibitors

in both pre-clinical and clinical settings. It will also be interesting to assess whether MAPK4 expression/activity is induced in TNBC that gain resistance to PI3K inhibition and whether targeting MAPK4 will overcome this acquired resistance.

We previously demonstrated that AKT activation is essential for mediating the tumor growth-promoting effects of MAPK4[16]. Our current data further confirmed the indispensable role of AKT activation in mediating MAPK4 activity in promoting TNBC growth. This leads to the question of whether MAPK4-high expression status, independent of *PIK3CA/PTEN/AKT1* alteration, will define TNBC tumors with a better response to AKT blockade. This is particularly important since a recent phase 3 IPATunity130 trial testing the AKT inhibitor ipatasertib (GDC-0068) plus paclitaxel in patients with PIK3CA/AKT1/PTEN-altered TNBC did not significantly improve progression-free survival when compared with placebo plus paclitaxel treatment group[28]. We are actively pursuing this research direction.

## Methods

**Reagents and antibodies**. The antibodies against p-AKT T308 (Catalog 13038), p-AKT S473 (Catalog 4060), AKT (Catalog 9272), GSK3β (Catalog 9315), and p-GSK3β S9 (Catalog 9336) were from Cell Signaling Technology. Other antibodies used include Anti-DYKDDDDK (Agilent, Catalog 200474), anti-MAPK4 (Abcepta, Catalog AP7298b), anti-BrdU antibody (MilliporeSigma, Catalog B2531), and anti-β-ACTIN (Abclonal, Catalog AC026 or MilliporeSigma, Catalog A1978). The kinase inhibitors used include PI3K inhibitors LY294002 (MilliporeSigma, Catalog L9908), Alpelisib/BYL-719 (MedChemExpress, Catalog HY-15244), Pictilisib/GDC-0941 (Selleckchem, Catalog S1065), and AKT inhibitors MK2206 and GSK2141795 (Selleckchem, Catalog S1078 and S7492). LipoD293 (Catalog SL100668) was purchased from SignaGen.

**Plasmids**. LentiCRISPR v2 (Addgene plasmid 52961) was a gift from Feng Zhang at MIT, Cambridge, Massachusetts. The pInducer10 vector was provided by Thomas Westbrook at Baylor College of Medicine, Houston, Texas. The pInducer20-YF vector and the pInducer20-YF and pInducer10 based constructs with Dox-inducible overexpression or knockdown of MAPK4 were described previously[16].

**Cell culture, transfection, lentivirus infection**. Cancer cell lines were obtained from the American Type Culture Collection (ATCC). For lentivirus-mediated gene delivery, lentiviral constructs were transfected into 293FT cells (Thermo fisher, Catalog R70007) using the LipoD293 transfection reagent (SignaGen, Catalog SL100668) together with the packaging mix of vectors pMD2.G and psPAX2. Viruses were harvested and applied for cell infection as described before[16]. The established cell lines were then expanded and stocked for further assays. The pInducer10 based constructs were used for lentivirus-mediated Dox-inducible knockdown of MAPK4 (iG2 and iG4) or control (iNT). The cells were induced with up to 4 μg/ml Dox for at least 3 days to obtain significant knockdown of MAPK4. The pInducer20-YF based constructs were used for lentivirus-mediated Dox-inducible overexpression of MAPK4 (iMAPK4) or control (iCtrl). The cells were treated with up to 1 μg/ml Dox for at least 2–3 days for ectopic overexpression of MAPK4.

The *MAPK4*-knockout (KO) SUM159 and MDA-MB-231 cell lines were created using a similar protocol as we described previously[16]. Knockout of *MAPK4* in each single clone was verified by genomic sequencing and western blots. The pCDH based lentiviral constructs were used for lentivirus-mediated stable overexpression of MAPK4 in the *MAPK4*-KO cell as described before[16].

**Western blot**. Cell lysates were prepared in RIPA buffer and protein concentrations were quantified using a Pierce BCA protein assay kit. An equal amount of protein (5–20 μg) was used in western blot analysis.

**Cell proliferation assays**. We used three approaches to assess cell proliferation, including direct cell counting, crystal violet staining–based cell proliferation assay, and BrdU incorporation assay as previously described[16]. When applicable, the kinase inhibitor(s) or vehicle control was added during the initial setup and replenished in fresh media every 3 days.

**Colony formation assay**. 1000–2500 single cells were suspended and seeded into each well of 6- or 12-well plates. The cells were then treated with the indicated inhibitors or vehicle control in triplicates for 10–21 days. Cells were then fixed with 10% (w/v) formaldehyde for 15 min and stained with 0.05% (w/v) crystal violet supplemented with 10% ethanol and 10% methanol for 20 min at room temperature. After a final wash of three times with distilled water, the plates were air-dried and scanned using a Canon scanner. The cell colonies were quantified either using ImageJ (area%) or by directly measuring absorbance (570 nm) of the solved crystal violet in 10% acetic acid. In the latter case, a background reading of 0.10 (the average OD reading of stained empty wells from multiple independent experiments using the same protocol) was reduced from all data points to remove background noises.

**Soft-agar colony formation assay**. Soft-agar colony formation assays were performed as described before[16]. 4 μg/ml or up to 1 μg/ml Dox were used for inducing knockdown or overexpression of MAPK4 in the indicated engineered cells. When applicable, PI3K inhibitors LY294002, Alpelisib, or Pictilisib at the indicated concentrations were added during the initial setup and replenished in fresh media every week. The colony numbers were counted and quantified after 3–4 weeks.

**Mammosphere formation assay**. A quantity of $1 \times 10^4$ single cells in 2 ml phenol red-free DMEM/F12 (Gibco, 21041025) containing B27 supplement (no vitamin A; Invitrogen, 12587) and SingleQuot™ (Lonza, 11645500) were added into each well of ultralow attachment 6-well plates pre-coated with Polyhydroxyethylmethacrylate (pHEMA). Five to ten days later, spheres were imaged under a microscope.

**Scratch wound healing assay**. Scratch wound healing assays were performed to evaluate cell migration on 6-well plates (100,000 cells/well). Cell monolayers at confluency were scratched with a 200 μl tip. Wound closure was analyzed at 8 and 24 h after scratch.

**Xenograft tumor models**. Female SCID/beige mice at 8–10 weeks old from Envigo were used in the xenograft studies. Mice were housed in a pathogen-free facility at Baylor College of Medicine. MDA-MB-231 and HCC1937 cells $(2 \times 10^6)$ with Dox-inducible knockdown of MAPK4 (ishMAPK4) vs. control (iNT) were injected into mammary fat pads (iNT, left side; ishMAPK4, right side). SUM159 cells with Dox-inducible expression of MAPK4 (iMAPK4) or control (iCtrl) were similarly injected. Mice began receiving 4 mg/ml (for inducible knockdown) or 0.5 mg/ml (for inducible overexpression) Dox in 1–10% sucrose in drinking water on the day of tumor injection and throughout the studies. Tumors were monitored/measured every week, and tumor volumes were calculated as $Vol = 0.52 \times abc$ (a, b, c: the maximum length of each dimension of the tumor). The wild type and *MAPK4*-KO MDA-MB-231 cells $(1 \times 10^6)$ were similarly used in the Alpelisib treatment studies. When tumors reach significant sizes (average of 100–200 mm³), Alpelisib (20 mg/kg) was delivered to mice daily through oral gavage. Tumors were similarly monitored, and tumor volume calculated every week. All tumors were harvested as indicated and weighed. Average tumor weight was compared among different groups for statistical relevance using the unpaired two-tailed Student's *t* test. $P < 0.05$ is considered statistically significant.

**Study approval**. All animal studies were approved by the Institutional Animal Care and Use Committee of Baylor College of Medicine.

**Statistics**. TCGA RNA-seq data were obtained from the Broad Institute's Firehose data portal (https://gdac.broadinstitute.org). P value was calculated by two-sided *t* test on log2-transformed expression values. The statistical relevance in the cell-culture studies and xenograft tumor studies was analyzed using the unpaired two-tailed Student's *t* test. When multiple comparisons were made, one-way or two-way ANOVA followed by Dunnett's multiple comparisons test or Sidak's multiple comparisons test was performed using GraphPad Prism, 9.3. $P < 0.05$ was considered significant.

**Reporting summary**. Further information on research design is available in the Nature Research Reporting Summary linked to this article.

## Data availability

Source data are provided with this paper.

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

## Acknowledgements

We thank Dr. Jeffrey M. Rosen for critically reading the paper and Fengjun Chen for technical support. This research was supported by grants from the Department of Defense Congressionally Directed Medical Research Programs (W81XWH-17-1-0043 to F.Y.), the Cancer Prevention Research Institute of Texas (RP130651, RP200493 to F.Y., and RP170691 to M.T.L.), the National Institutes of Health (CA125123 to C.J.C., U54-CA224076 and CA125123 to M.T.L.), and the R.P. Doherty Jr.–Welch Chair in Science (Q-0022 to D.D.M.). We also acknowledge the joint participation by Adrienne Helis Malvin Medical Research Foundation through its direct engagement in the continuous active conduct of medical research in conjunction with Baylor College of Medicine and the MAPK4 as a Novel Therapeutic Target for Human Cancers Cancer Program.

## Author contributions

F.Y. conceived the ideas, W.W., D.H., Q.C., T.S., B.D., and F.Y. designed the experiments, W.W., D.H., Q.C., B.D., M.T.L., P.Y., C.J.C., D.D.M., and F.Y. wrote and revised the manuscript, W.W., D.H., Q.C., T.S., R.W., Y.M., and W.Z. performed the experiments, and W.W., D.H., Q.C., B.D., M.T.L., P.Y., C.J.C., D.D.M., and F.Y. analyzed the data.

## Competing interests

M.T.L. is founder and limited partner in StemMed Ltd, and founder and manager in StemMed Holdings, its general partner. He also holds an equity stake in Tvardi Therapeutics Inc. The other authors declare no competing interests.

## Additional information

**Peer review information** *Nature Communications* thanks Qiang Sheng and the other anonymous reviewer(s) for their contribution to the peer review this work. Peer reviewer reports are available.

