## [Peer Review File · Nature Communications]

Reviewers' Comments:

Reviewer #1:

Remarks to the Author:

This manuscript by Wang et al describes that MAPK4, which is overexpressed in 30% of basal-like breast cancer cases, promotes cell growth, activates AKT and associates with responses to p110 inhibitors of TNBC cells. The findings that MAPK4 is an oncogene and that MAPK4 causes p110-independent AKT activation are extension of the authors' previous publication, albeit in other cancer types (prostate, colon and lung). What had not been addressed previously was the therapeutic influence of the overexpression. However, this part is not sufficiently addressed in the manuscript and is limited to studies using cell lines.

More specific comments-

Is there correlation between MAPK4 protein levels and sensitivity to p110 inhibitors? The authors mentioned that SUM159 cells (which have relatively lower MAPK4 level) responded better to the inhibitors. But the data presented are not convincing and the sample size is too small. Also, very importantly, all the drug response assays were performed with single dose. It'd be more informative if one performs serial dilution assays and includes higher doses of the inhibitors. 1uM was used in the experiments and there is certainly room for increment. Are the differential sensitivities still observed in high doses? And whether the so called "resistant" cells before silencing MAPK4 achieve IC50? Are they "real" resistant cells.

MAPK-overexpressing cells have AKT activated, did the author try to inhibit AKT or mTOR? The authors suggest that inhibition of MAPK4 sensitize cells to p110 inhibitors. Without an MAPK4-specific inhibitor available, the clinical feasibility of this strategy is questionable.

What is the mutation status of the cell lines used?

Are the effects of MAPK4 specific to TNBC? Note that there are several TCGA luminal A cases which had comparably high MAPK4 levels. The observations that normal breast MCF10A cells responded similarly to MAPK4 overexpression as TNBC cells further argue the specificity.

Although MAPK4 overexpression is observed in certain population of TNBC, whether the whole MAPK4-overexpressing population have the same manifestation, i.e. promoted tumor growth and reduced p110 inhibitor response due to MAPK4, is not certain. The title of the manuscript shall be revised.

Reviewer #2:

Remarks to the Author:

The manuscript by Wang W et al entitled "Targeting MAPK4 in a large subset of triple-negative breast cancer represses 2 tumor growth and resistance to PI3K blockade" described a study in which MAPK4 was found upregulated in a large cohort of TNBC patients. Further, activated MAPK4 promotes mammary carcinogenesis, whereas suppressed MAPK4 inhibits AKT activation, cell proliferation and anchorage-independent growth in TNBC cells and xenograft growth in vivo. Interestingly, activation of MAPK4 confers resistance towards PI3K inhibitors by activating AKT signaling. These intriguing and important findings establish a role for MAPK4 as a driver for TNBC progression, and as a promising target for combination therapy with PI3K inhibitors. The study was well designed and executed, with solid assessments at various molecular, cellular, organ, whole animal levels and human samples and publicly available databases, with multiple models to support their hypothesis and conclusion. This study provides evidence and a rationale to translate into future clinical implications and applications. A number of major and minor concerns are raised here for the authors to improve the overall quality of the manuscript before acceptance.

Major concerns:

1. TNBC is prone to develop metastasis, if MAPK4 is a key driver for TNBC development and progression, a role of MAPK4 in driving metastasis is assumed. However, there is a lack of literature summary in introduction/discussion, or relevant assessments on cellular mobility of

TNBC cells and in vivo models.

2. Fig 1B, any matching MAPK4 protein level data for the 56 BCa PDX models?
3. Fig 4A-C and Fig 7C need to demonstrate changes of protein level of MAPK4 and biomarkers of proliferation, apoptosis, motility, and downstream genes for MAPK4.
4. The authors should discuss potential implications of this study in association with standard chemotherapy, recurrence, and metastasis of TNBC.

Minor Concerns:

1. Legend for Fig 3(K) should be 3(J).

Response to Reviewer #1

Critiques #1: Is there correlation between MAPK4 protein levels and sensitivity to p110 inhibitors? The authors mentioned that SUM159 cells (which have relatively lower MAPK4 level) responded better to the inhibitors. But the data presented are not convincing and the sample size is too small. Also, very importantly, all the drug response assays were performed with single dose. It'd be more informative if one performs serial dilution assays and includes higher doses of the inhibitors. 1uM was used in the experiments and there is certainly room for increment. Are the differential sensitivities still observed in high doses? And whether the so called "resistant" cells before silencing MAPK4 achieve IC50? Are they "real" resistant cells.

Response: We thank Reviewer #1 for raising these important questions. We agree that our sample size is too small to claim a broad correlation between MAPK4 protein levels and sensitivity to p110 inhibitors. As described in new Figures 7, 8, and part of Figure 9, we have generated additional engineered TNBC cell lines, performed serial dilution assays, and included higher doses of the PI3K inhibitors. Altogether, our data on MAPK4 overexpression/knockdown/knockout in a total of n=7 commonly used human TNBC cell lines demonstrated that the TNBC cells with higher MAPK4 protein expression (either the control cells in the knockdown/knockout studies or the experimental cells in the overexpression studies) consistently exhibited higher growth and resistance to PI3K blockade when compared with those isogenic cells with lower MAPK4 expression (the corresponding experimental cells in knockdown/knockout studies or the control cells in overexpression studies). Therefore, there is a clear correlation between MAPK4 protein levels and sensitivity to p110 inhibitors within the isogenic cells. We have modified our manuscript accordingly. Our current study provides strong evidence for MAPK4 overexpression promoting and MAPK4 inhibition repressing TNBC cell/tumor growth and resistance to PI3K blockade. Finally, the MAPK4 expression in the MAPK4-high TNBC tumors does create a therapeutic opportunity to target MAPK4 to both repress tumor growth and sensitize them to PI3K blockade.

Critiques #2: MAPK4-overexpressing cells have AKT activated, did the author try to inhibit AKT or mTOR?

Response: We thank Reviewer #1 for this question, which is also specifically highlighted by the editor. Please find our answer above in our Response #3 to Editor.

Critiques 3: The authors suggest that inhibition of MAPK4 sensitize cells to p110 inhibitors. Without an MAPK4-specific inhibitor available, the clinical feasibility of this strategy is questionable.

Response: We thank Reviewer #1 for pointing out this problem. To our knowledge, we are one of the few groups studying MAPK4 and the first to uncover its critical roles in human cancers. We expect that the field will realize the importance of targeting MAPK4 in human cancers and make significant efforts to develop the first-ever MAPK4-specific inhibitor. In fact, we are starting significant efforts to screen for the first MAPK4-specific inhibitor. Finally, as discussed in Response #3 to Editor, one of our ongoing studies is to fully address the effectiveness of targeting AKT in the MAPK4-high TNBC. We will test our hypothesis that MAPK4-high status, rather than the *PIK3CA/PTEN/AKT1* alteration status that failed in the recent phase 3 IPATunity130 trial, will define TNBC tumors with better response to AKT blockade (<https://www.cpr.it.state.tx.us/grants-funded/grants/rp200439>). We hope that the clinical relevance of our efforts will be established in the near future.

Critiques 4: What is the mutation status of the cell lines used?

Response: We thank Reviewer #1 for this important question. Please refer to the table below for the

PIK3CA mutation status of the TNBC cell lines used. We also included a brief discussion on this as following.

“Interestingly, five out of the seven TNBC cell lines that we have tested are either *PTEN*-null (HCC1395, HCC1937, and MDA-MB-468 cells with frameshift mutation or homodeletion of *PTEN*) or *PIK3CA* (SUM159) or *PIK3R1* mutated (HS578T). We are investigating the detailed molecular mechanism underlying the maintenance of the MAPK4-addiction phenotype in these TNBC cells.”

	HCC1395	HS578T	HCC1806	HCC1937	MDA-MB-231	MDA-MB-468	SUM159
PIK3CA mutation	WT	WT (PIK3R1 mut)	WT	WT	WT	WT	H1047L
PTEN mutation	frameshift/null	WT	WT	null	WT	frameshift/null	WT
PTEN protein	-	+	+	-	+	-	+

Critiques 5: Are the effects of MAPK4 specific to TNBC? Note that there are several TCGA luminal A cases which had comparably high MAPK4 levels. The observations that normal breast MCF10A cells responded similarly to MAPK4 overexpression as TNBC cells further argue the specificity.

Response: As described in our response #2 to Editor, we do not believe that the effects of MAPK4 are specific to TNBC. However, what makes targeting MAPK4 in TNBC unique and potentially highly important is that a large subset of TNBC (30-50% or more) highly express MAPK4. In contrast, the frequency of MAPK4-high expression in many other cancer types, including BCa subtypes such as the luminal A pointed by Reviewer #1, is low (frequently less than 5%). Finally, unlike ER+ or HER2+ types of BCa, there is not yet an approved targeted therapy (other than immunotherapy) for TNBC, which also makes targeting MAPK4 in TNBC very important.

Critiques 6: Although MAPK4 overexpression is observed in certain population of TNBC, whether the whole MAPK4-overexpressing population have the same manifestation, i.e. promoted tumor growth and reduced p110 inhibitor response due to MAPK4, is not certain. The title of the manuscript shall be revised.

Response: Following Reviewer #1’s suggestion, we are submitting the revised manuscript under the title of “MAPK4 expression in a large subset of triple-negative breast cancer promotes tumor growth and resistance to PI3K blockade.” Another candidate title can be “MAPK4 promotes triple-negative breast cancer tumor growth and resistance to PI3K blockade.”

Response to Reviewer #2

Critique 1: TNBC is prone to develop metastasis, if MAPK4 is a key driver for TNBC development and progression, a role of MAPK4 in driving metastasis is assumed. However, there is a lack of literature summary in introduction/discussion, or relevant assessments on cellular mobility of TNBC cells and in vivo models.

Response: We thank Reviewer #2 for this important question. We assessed MAPK4 effects on TNBC cellular mobility using wound healing assays on four representative TNBC cell lines, including MDA-MB-231, HCC1937, SUM159, and HCC1806 cells. Knockdown of MAPK4 in the MAPK4-high HCC1937 cells greatly inhibited its migration; in contrast, neither knockdown of MAPK4 in the MAPK4-high MDA-MB-231 cells nor overexpression of MAPK4 in the MAPK4-medium/low SUM159 or HCC1806 cells

significantly affected their migration (New Supplementary Figure S1). Therefore, MAPK4 only promotes cell mobility in 1 out of 4 TNBC cell lines that we have tested. Based on these data, we concluded that, unlike the growth-promoting activity, MAPK4's ability to regulate (promote) cell motility appears cell-context dependent. We also included the following discussion on this.

"In contrast, MAPK4 promoted cell migration in the wound healing assay in only one out of four TNBC cell lines tested (Supplementary Figure S1). Therefore, unlike the growth-promoting activity, MAPK4's ability to promote cell motility appears cell context dependent."

Critique 2: Fig 1B, any matching MAPK4 protein level data for the 56 BCa PDX models?

Response: We have examined the expression of MAPK4 protein (Western blots) in some of these TNBC PDX. We confirmed the correlation of MAPK4 mRNA expression with protein expression in many cases. However, as expected, there are also cases that MAPK4 mRNA did not correlate with MAPK4 protein expression (such as the MAPK4 mRNA-high BCM-4913 and BCM-6257 PDX showed relatively low expression of MAPK4 protein, and vice versa the MAPK4 mRNA-low MC1 exhibited high expression of MAPK4 protein). A representative blot is shown in *Data to reviewer, Panel C*.

Critique 3: Fig 4A-C and Fig 7C need to demonstrate changes of protein level of MAPK4 and biomarkers of proliferation, apoptosis, motility, and downstream genes for MAPK4.

Response: Since we have not identified a MAPK4 antibody suitable for IHC staining, we relied on Western blots to assess MAPK4 and its downstream AKT phosphorylation in the tumors. We confirmed that MAPK4 overexpression in SUM159 xenografts led to enhanced AKT phosphorylation (Gain-of-function studies, *Data to reviewer, Panel D*). We previously reported that knockdown of MAPK4 H157 xenograft tumors led to repressed AKT phosphorylation, inhibited proliferation, and enhanced apoptosis (Supplementary Figure S1 in Wang et al., JCI, 2019). However, these effects were only consistently observed in tumors after short-term repression of MAPK4 (such as after 1-week Dox-induced knockdown of MAPK4 in previously established non-induced H157-ishMAPK4 tumors, Wang et al., JCI, 2019). In our current study, the Dox-induced HCC1937-ishMAPK4 cell injections did not produce any tumors (Figure 4B). The tumors that eventually grew in the other MAPK4-repressed groups (Figure 4A, 10C) are expected to be resistant to "MAPK4" inhibition and/or Alpelisib treatment with less consistent MAPK4-blockage induced events. Finally, we performed further in vitro studies showing that MAPK4 regulates AKT phosphorylation and apoptosis in TNBC cell lines with knockout of *MAPK4* or Dox-induced expression of MAPK4, and these effects were further regulated by PI3K inhibitors Alpelisib and Pictilisib (*Data to Reviewer, Panel E*). However, we did not detect consistent MAPK4 induction of the cell proliferation gene Cyclin D1, although *MAPK4*-KO significantly repressed its expression in SUM159 cells. We are performing detailed studies on the molecular mechanism underlying MAPK4 regulation of apoptosis. Finally, since we did not observe consistent MAPK4 regulation of cell motility (Supplementary Figure S1), we did not further investigate MAPK4 regulation of biomarkers of motility.

Critique 4: The authors should discuss potential implications of this study in association with standard chemotherapy, recurrence, and metastasis of TNBC.

Response: We thank Reviewer #2 for this wonderful suggestion. We have included the discussions on potential implications of this study in association with standard chemotherapy (page 12, line 21 and in recurrence, such as that resistance to PI3K inhibition (page 14, line 1). Our current data only partially support a potential context-dependent role of MAPK4 in promoting cell migration; however, by driving primary tumor mass growth, it is possible that MAPK4 can more universally drive metastasis of TNBC secondary to the primary tumor growth.

Critique 5: Legend for Fig 3(K) should be 3(J).

Response: We thank Reviewer #2 for pointing out this error. We have updated this in the revised manuscript.

Reviewers' Comments:

Reviewer #1:

Remarks to the Author:

In this revision, the authors provided more data related to the therapeutic indication of MAPK levels. The new data repetitively demonstrated that MAPK4 knockdown increased sensitivity to PI3K inhibitors, whereas MAPK4 overexpression led to resistance. Some questions still remain.

First, it is my original concern regarding the use of the cell models to study PI3K inhibitor resistance. My original comment said "And whether the so called "resistant" cells before silencing MAPK4 achieve IC50? Are they "real" resistant cells."

The authors did not reply to this.

Specifically, whether endogenous MAPK4 levels across cell lines/cell models correlate with PI3K inhibitor response. And whether knockdown MAPK reverses the resistance of the "real" resistant cells. From the new data provided, all the control cell lines were inhibited >50% in the experiments, regardless of the endogenous MAPK4 levels.

Second, with the new Figure 7 data, the authors stated that knockdown of MAPK4 sensitized the cells to Pictilisib and Alpelisib treatments. The way that the data are currently presented cannot show whether the knockdown cells were indeed more sensitive to the inhibitors. Although the OD readings of knockdown cells were lower than that of control cells under the same inhibitor doses, the knockdown alone already caused reduction in OD readings (i.e. compare the readings at 0uM of iNT, iG2 and iG4).

Figure 8, likewise, since overexpression of MAPK4 alone promoted tumor growth and in turn the OD readings, whether the cells were more resistant to the inhibitors by MAPK4 is not clearly presented.

Also, with the OD readings, the colony pictures on the left panel are redundant and not very useful.

Third, the new Figure 6 data showed that inhibition of AKT by AKT inhibitors abolished the MAPK4 activities in enhancing the growth of MAPK4-overexpressing cells. Akt is downstream of PI3K. What would be the possible explanation of cells being resistant to PI3K inhibitor but sensitive to AKT inhibitor? What is the status of AKT activation in MAPK4-overexpressing cells treated by the PI3K inhibitors?

Others:

Fig.1B – the authors included these new data of PDX and stated in the figure legend "MAPK4 gene expression percentile is at around 50% of all genes expressed in these PDX tumors". Please clarify whether a 50%-percentile represents high level of expression?

Line 131- This sentence, which refers to the in vitro proliferation experiment, says "MAPK4 knockdown in MDA-MB-231 cells did not significantly affect their proliferation (data not shown)." Please add the data in, especially MDA-MB-231 was also used in mice experiments in which MAPK4 knockdown inhibited tumor growth.

In the new Figure 7F, as opposed to the claim that MAPK4 knockdown decreases p-AKT (T308, S473), pAKT levels were increased in this experiment? Data of the same cell line was shown in Figure 2B, in which phosphorylated AKT levels were decreased in MAPK knockdown cells.

In the Western blot data, there is no total level of GSK3beta.

The newly proposed title is still very similar to the previous one.

Reviewer #2:

Remarks to the Author:

The authors have adequately addressed my comments.

Response to Reviewer #1

Critique #1: It is my original concern regarding the use of the cell models to study PI3K inhibitor resistance. My original comment said “And whether the so called “resistant” cells before silencing MAPK4 achieve IC50? Are they “real” resistant cells.” The authors did not reply to this. Specifically, whether endogenous MAPK4 levels across cell lines/cell models correlate with PI3K inhibitor response. And whether knockdown MAPK4 reverses the resistance of the “real” resistant cells. From the new data provided, all the control cell lines were inhibited >50% in the experiments, regardless of the endogenous MAPK4 levels.

Response: We agree with Reviewer #1 on his/her concern about the “resistant” status of the cells we are using. Because of the limited evidence for endogenous resistance of TNBC cell lines such as MDA-MB-231 etc., to PI3K blockade, we have removed the term “resistance” or “resistant” from our main text. Instead, we now only claim that MAPK4 regulates TNBC cell/xenograft “sensitivity” to PI3K blockade in the Result section, which more concisely describes our data. We thank Reviewer #1 for pointing this out.

As discussed in our previous response to Reviewer #1, we can't claim a broad correlation between MAPK4 protein levels and cell sensitivity to p110 inhibitors among different cell lines. This is partially due to the limited number (n=7) of cell lines examined and the complex genetic makeup of different TNBC cell lines, which represent a diverse collection of advanced BCa types. A large number of TNBC cell lines, PDX models, or patients in the clinic will be needed to assess the correlation between MAPK4 protein expression and tumor response to PI3K blockade. Therefore, we have focused our current efforts on the correlation between MAPK4 protein expression and tumor response to PI3K blockade in otherwise isogenic cell lines, as discussed below.

As described in Figures 7-9 and the new Supplementary Figure S1, our data on MAPK4 overexpression/knockdown/knockout in a total of n=7 commonly used human TNBC cell lines demonstrate that TNBC cells with higher MAPK4 protein expression (either the control cells in the knockdown/knockout studies or the experimental cells in the overexpression studies) consistently exhibit higher growth and less sensitivity to PI3K blockade when compared with otherwise isogenic cells with lower MAPK4 expression (the corresponding experimental cells in knockdown/knockout studies or the control cells in overexpression studies). Therefore, there is a clear correlation between MAPK4 protein levels and cell sensitivity to p110 inhibitors within the isogenic cells.

Collectively, our data provide strong evidence that MAPK4 regulates TNBC cell/tumor growth and sensitivity to PI3K blockade. Thus, targeting MAPK4 in at least a subset of high MAPK4 expressing TNBC tumors is an important new therapeutic strategy to both repress tumor growth and sensitize to PI3K blockade.

Critique #2: With the new Figure 7 data, the authors stated that knockdown of MAPK4 sensitized the cells to Pictilisib and Alpelisib treatments. The way that the data are currently presented cannot show whether the knockdown cells were indeed more sensitive to the inhibitors. Although the OD readings of knockdown cells were lower than that of control cells under the same inhibitor doses, the knockdown alone already caused reduction in OD readings (i.e. compare the readings at 0uM of iNT, iG2 and iG4). Figure 8, likewise, since overexpression of MAPK4 alone promoted tumor growth and in turn the OD readings, whether the cells were more resistant to the inhibitors by MAPK4 is not clearly presented. Also, with the OD readings, the colony pictures on the left panel are redundant and not very useful.

Response: We have reorganized and modified the data in Figure 7 and Figure 8. The Y-axis scales were adjusted to maximize the visualization of different data points. And as described in the Methods section, we also reduced 0.10 (the average OD reading of stained empty wells from multiple independent experiments using the same protocol) from all data points to remove background noises. We include

colony photos in Figures 7 and 8 so that readers have a direct visualization of the difference in colony numbers and sizes. Finally, we can move them to a new Supplementary Figure if needed.

We believe that the Figure 7 and Figure 8 data provide evidence supporting MAPK4 regulation of TNBC cell sensitivity to PI3K blockade. For example, in Figure 7B, while neither Pictilisib nor Alpelisib (at 0.5-1 μ M) inhibited SUM159-iNT cell growth, they both significantly reduced SUM159-iG2 and SUM1159-iG4 cell growth under the same treatments. Finally, in the new Supplementary Figure S1B-C, we include a modified version of data quantification in Figures 7-8. The vehicle treated arms are all set at 100%, which provides another angle to visualize and directly compare the different responses of MAPK4-modified TNBC cells to PI3K inhibitor treatments.

Critique #3: The new Figure 6 data showed that inhibition of AKT by AKT inhibitors abolished the MAPK4 activities in enhancing the growth of MAPK4-overexpressing cells. Akt is downstream of PI3K. What would be the possible explanation of cells being resistant to PI3K inhibitor but sensitive to AKT inhibitor? What is the status of AKT activation in MAPK4-overexpressing cells treated by the PI3K inhibitors?

Response: A central aspect of our results is that both MAPK4 and PI3K pathways can activate AKT. Inhibiting PI3K alone leaves the MAPK4-AKT signaling axis largely intact, providing an alternate pathway for MAPK4 to regulate TNBC cell sensitivity to PI3K blockade. In contrast, AKT functions downstream of both MAPK4 and PI3K pathways as a central node for tumor promotion. Therefore, AKT inhibitors abolished the ability of MAPK4 to enhance growth of MAPK4-overexpressing cells. Finally, as shown in the new Figure 8E, AKT phosphorylation and activities (phosphorylation of GSK3beta) are partially maintained in the MAPK4-overexpressing cells treated by the PI3K inhibitor.

Critiques 4: Fig. 1B – the authors included these new data of PDX and stated in the figure legend “MAPK4 gene expression percentile is at around 50% of all genes expressed in these PDX tumors”. Please clarify whether a 50%-percentile represents high level of expression?

Response: The 50%-percentile criterion represents the expression of MAPK4 relative to all other genes. This means that MAPK4 expression level is significant in these TNBC PDXs (higher than almost 50% of all genes). We have made the following statement “*MAPK4 is markedly expressed (at around the 50th percentile of all genes expressed) in these PDX tumors (Panel B)*” in Figure 1 legend.

Critiques 5: Line 131- This sentence, which refers to the in vitro proliferation experiment, says “MAPK4 knockdown in MDA-MB-231 cells did not significantly affect their proliferation (data not shown).” Please add the data in, especially MDA-MB-231 was also used in mice experiments in which MAPK4 knockdown inhibited tumor growth.

Response: We showed this data in Figure 3C in our original submission. We removed this data to make room for the new data for the first revision. We now add this data back in the new Supplementary Figure S1A in this second revision.

Critiques 6: In the new Figure 7F, as opposed to the claim that MAPK4 knockdown decreases p-AKT (T308, S473), pAKT levels were increased in this experiment? Data of the same cell line was shown in Figure 2B, in which phosphorylated AKT levels were decreased in MAPK knockdown cells.

Response: The Figure 2B data was produced using cells under regular culture conditions. In contrast, Figure 7F data was generated using cells ten days after PI3K inhibitor vs. control treatment in the clonogenic assay settings. The cells were seeded at low density (clonogenic assay) to allow clonal

outgrowth of single cells. Accordingly, only the most “fit” cells in the MAPK4-knockdown group will grow into colonies. Apparently, these most fit cells somehow upregulate PI3K/PDK1 pathway to phosphorylate/activate AKT. We also repeated these data using the MAPK4-knockdown SUM159 cells an additional three times and generated new data using the MAPK4-knockdown MDA-MB-231 cells (also three repeats). We consistently observed that the MAPK4-knockdown SUM159 and MDA-MB-231 cells partially regained AKT phosphorylation/activation in these clonogenic assays (reasons as discussed above). However, during these repeats, we realized that our previous data on the seemingly higher phosphorylation of AKT in the MAPK4-knockdown SUM159 cells were likely based on experimental variation. Hence, we have replaced those data using updated complete data sets also incorporating total GSK3beta protein expression (to address Critique 7 below as well).

Finally, we described these data as *“Interestingly, after ten days culture in the clonogenic assay settings (cells plated at low density for clonal growth of individual cells), the MAPK4-knockdown SUM159 and MDA-MB-231 cells partially regained AKT phosphorylation/activation, presumably due to PI3K-AKT pathway activation for individual cell growth/survival. In accord with this, AKT phosphorylation/activation in these MAPK4-knockdown cells was highly sensitive to PI3K blockade (0.5 μM Alpelisib), further confirming MAPK4-AKT as the essential pathway for regulating TNBC sensitivity to PI3K inhibition (Figure 7F).”*

Critique 7: In the Western blot data, there is no total level of GSK3beta.

Response: We now include the total GSK3beta protein loading controls in the Western blots in Figure 2C (experiments performed on the same cell lysates for the other readouts there). As expected, there is no change in total protein levels of GSK3beta. We also confirmed a lack of GSK3beta protein level change in the newly generated Figure 7F and 8E Western blots (those on the PI3K inhibitor vs. control treated MAPK4-knockdown and MAPK4-overexpressing TNBC cells). All these data support that MAPK4 (both gain and loss of function) only affects GSK3beta phosphorylation, but not GSK3beta protein levels.

Critique 8: The newly proposed title is still very similar to the previous one.

Response: We propose the following new title, and we welcome Reviewer #1’s suggestion.

“MAPK4, highly expressed in a large subset of triple-negative breast cancer, promotes TNBC growth and reduces tumor sensitivity to PI3K blockade.”

Reviewers' Comments:

Reviewer #1:

Remarks to the Author:

I have no further comments.

Response to Reviewer #1

Comment #1: I have no further comments.

Response: We thank Reviewer #1 for his comment. We also thank the editorial decision for the acceptance of our manuscript.